# Highly efficient three-dimensional solar evaporator for high salinity desalination by localized crystallization

Lei Wu [1,6], Zhichao Dong [2,6], Zheren Cai[1,3], Turga Ganapathy[4], Niocholas X. Fang[4], Chuxin Li[2], Cunlong Yu[5], Yu Zhang[1,3] & Yanlin Song[1,3]*

Solar-driven water evaporation represents an environmentally benign method of water purification/desalination. However, the efficiency is limited by increased salt concentration and accumulation. Here, we propose an energy reutilizing strategy based on a bio-mimetic 3D structure. The spontaneously formed water film, with thickness inhomogeneity and temperature gradient, fully utilizes the input energy through Marangoni effect and results in localized salt crystallization. Solar-driven water evaporation rate of $2.63 \, kg \, m^{-2} \, h^{-1}$, with energy efficiency of >96% under one sun illumination and under high salinity (25 wt% NaCl), and water collecting rate of $1.72 \, kg \, m^{-2} \, h^{-1}$ are achieved in purifying natural seawater in a closed system. The crystalized salt freely stands on the 3D evaporator and can be easily removed. Additionally, energy efficiency and water evaporation are not influenced by salt accumulation thanks to an expanded water film inside the salt, indicating the potential for sustainable and practical applications.

[1] Key Laboratory of Green Printing, Institute of Chemistry, Chinese Academy of Sciences, Zhongguancun North First Street 2, Beijing 100190, PR China. [2] CAS Key Laboratory of Bio-inspired Materials and Interfacial Sciences, Technical Institute of Physics and Chemistry, Chinese Academy of Sciences, 29 Zhongguancun East Road, Beijing 100190, PR China. [3] University of Chinese Academy of Sciences, Beijing 100049, PR China. [4] Department of Mechanical Engineering, Massachusetts Institute of Technology, Cambridge, MA 02139, USA. [5] Beihang University, Xueyuan Road No. 37, Beijing 100191, PR China. [6] These authors contributed equally: Lei Wu, Zhichao Dong *email: ylsong@iccas.ac.cn

Facing the globally occurring water scarcity situation, solar-driven water evaporation or solar steam generation is considered as a promising technology for potential applications in desalination and clean water preparation[1–3], as solar energy is the only energy input to purify brine or polluted water. Increasing the energy efficiency from sunlight to water evaporation endotherm, decreasing heat loss and inhibiting pollutant or salt blockage during solar steam generation are critical factors both in fundamental research and further practical implementation of the solar-driven interfacial evaporation system[4]. Based on these factors, several strategies have been proposed to enhance water evaporation and energy efficiency through developing photothermal materials with effective absorption ability[5–9], minimizing heat loss and enhancing heat localization[10–14], providing effect water/steam transport interface or increasing energy output pathway[15–20] and increasing durability and antifouling property in extreme environment[21–24]. However, the effective utilization of input solar energy and converted heat under high salinity remains a challenge.

Recently, the employment of hierarchically nanostructured gels has been shown to reduce the water evaporation enthalpy and lead to a high water evaporation rate under one sun[25]. Salt blockage can be inhibited by manually introducing salt-rejection pathways. Considering the salt condensation or the heat loss, the efficiency is still limited[26,27]. The increased salt concentration during the continuous evaporation process inevitably results in the crystallization or accumulation of salt on the surface of photothermal materials, which will decrease effective light absorption, block the steam generation and inhibit the implementation of the concept of solar steam generation in real-world application[28,29]. Progresses have been made in recent years: the contactless design can separate the absorber from the water interface, resulting in the contamination resistance for a long time[30], and the edge crystallization can isolate the crystallized salt from the evaporator through gravity[31]. However, to improve the evaporation rate and energy efficiency for practical usage, there is still a long way to go. It is therefore worth further investigating how to increase the evaporation rate, energy efficiency and the sustainability of the photothermal materials under high salinity (Supplementary Table 1 and Supplementary Fig. 1a, b, which covers the current state in the solar desalination field with the variation of salinity).

To achieve effective water evaporation, efficient water transport and vapor evaporate route for efficient thermal management in the solar-driven evaporation process is critical. The steam generation rate and the energy efficiency have almost been pushed to the upper limit for the current solar steam generation system based on a water/structure interface with homogeneous film thickness and temperature without varying the water evaporation enthalpy[32,33]. Here, we report an efficient energy reutilizing strategy to achieve high energy efficiency and water evaporation rate under high salinity. The system is based on interfacial water film inhomogeneity management through hierarchical water pathways based on a biomimetic 3D structure prepared from size-dependent resin refilling induced additive manufacturing. Ascribing from the position-related structure inhomogeneity of the 3D structure, the generated water film on the evaporator surface displays a thickness gradient. In addition, the input energy acquired by the biomimetic 3D evaporator system is related to the distance between the light source and the precise position, which results in the position-related energy utilization of the illumination. Cooperating with the position-related water evaporation on the biomimetic 3D evaporator induced by the structure inhomogeneity, the liquid film displays temperature gradient along the liquid film, indicating that the whole system unevenly utilizes energy. The simultaneously formed Marangoni flow leads to the water supplementation to the more vigorous evaporation site to enhance evaporation and energy

efficiency, which finally leads to the energy reutilization property inside the water film. High water evaporation rate in darkness (1.17 kg m$^{-2}$ h$^{-1}$), high solar-driven water evaporation rate (2.63 kg m$^{-2}$ h$^{-1}$) and high energy efficiency (>96%) are achieved under one sun illumination with excellent stability even under high salinity. In addition, the liquid film thickness gradient and the position-related water evaporation along the liquid film also leads to the salt concentration gradient and localized salt crystallization feature on the 3D evaporator. Salt crystallizes at the apex liquid film on the structure with a thin layer of liquid in between, which endows the evaporator with salt free-standing and easy-removing property, proving its capability for sustainable utilization. Furthermore, the batch water purification rate in a closed system can reach 1.72 kg m$^{-2}$ h$^{-1}$ for continuously purifying the natural seawater sample, indicating its potential for practical applications in the future.

## Results

**Biomimetic design of the 3D evaporator.** We designed the solar evaporator structure inspired by the super liquid transportation property of the asymmetric capillary ratchet of the bird beak (Fig. 1a)[34] and the pitcher plant peristome surface (Fig. 1b)[35–37]. As shown in Fig. 1c, the asymmetric grooves and microcavity arrays with a dimensional gradient along each groove form the 3D structure with a height-to-diameter (H/D) ratio of 0.7 (Supplementary Fig. 2a). Size-dependent resin refilling induced additive manufacturing based on a self-made Digital Light Processing (DLP) continuous 3D printing system is employed to fabricate the 3D structures with surface distributed micropores[38] (Fig. 1d, Supplementary Note 1). The formation of composite resin is revealed in Supplementary Fig. 3a, where carbon nanotubes (CNTs, Supplementary Fig. 3b) and sodium citrate (Supplementary Fig. 3c) are added in the self-made UV curable resin. CNTs are chosen as the photothermal material[39] and sodium citrate particles[40] are employed as the surface distributed pore producer for the 3D evaporator. The sodium citrate particles are not able to flow along with the refilling resin during the continuous printing process, as the slicing thickness (5 μm) is much smaller than the particle dimension (Fig. 1d inset). Particles are solidified only on the surface of the cured structure instead. After removing the sodium citrate on the surface, micropores are thus achieved only on the 3D structure surface. Microcomputed tomography (Micro-CT) images shown in Fig. 1e–g characterize the inner and surface morphology of the biomimetic 3D evaporator, proving the successful preparation as designed. Scanning electron microscopy (SEM) further shows that the 3D evaporator exhibits randomly distributed micropores only on the surface (Fig. 1h–j). The composite plane film prepared from the same manufacturing process can absorb over 90% of input light (Supplementary Fig. 4), which is suitable as the material for preparing 3D evaporator.

**Generation and characterization of liquid film.** A high-speed camera is utilized to monitor the water movement on the biomimetic 3D evaporator surface (Fig. 2a–d). Time sequence images exhibit ultra-fast water spreading process (100 ms): water precursor moves upwardly along grooves of microcavity arrays and spreads perpendicularly to each groove simultaneously, forming a continuous liquid film covering the whole evaporator (Supplementary Movie 1). After the spreading process of 100 ms, the water/structure interface is generated. In contrast, without the surface distributed micropores on the evaporator surface (without the introduction of sodium citrate), the time needed for water/structure interface generation is increased to 2 s (Supplementary Fig. 5). Therefore, the asymmetric grooves and gradient microcavity arrays can facilitate the water suction from bulk water to

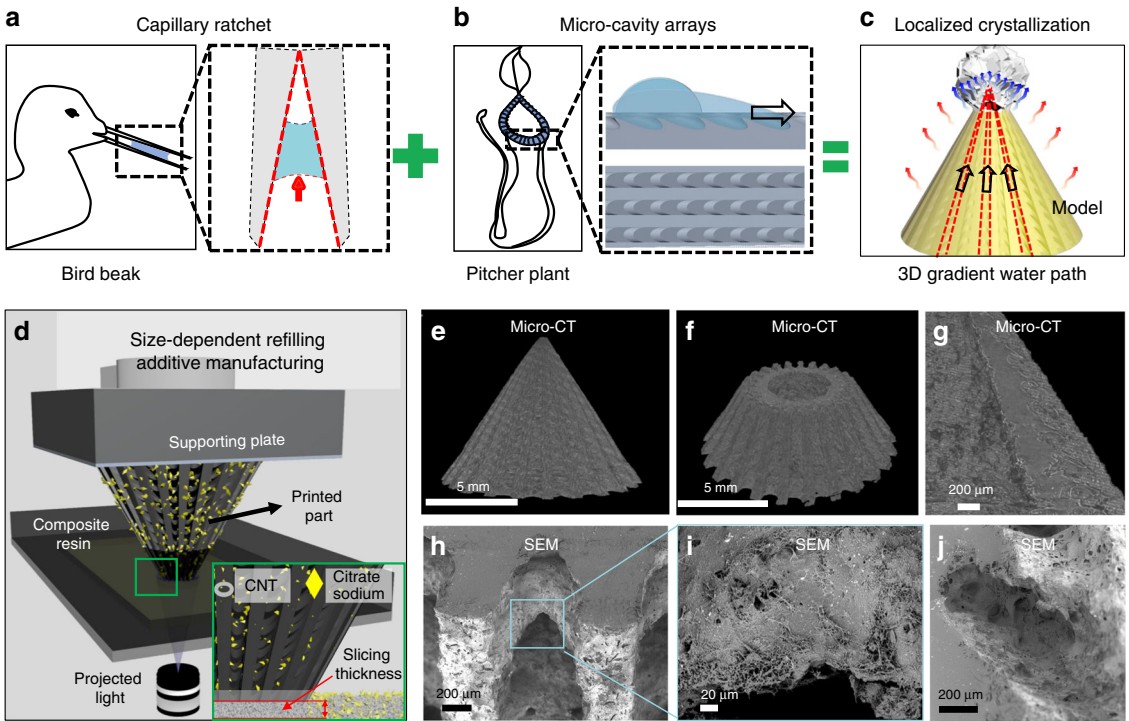

**Fig. 1 Additive manufacturing and characterization of the biomimetic 3D solar evaporator. a–c** Design of the biomimetic 3D evaporator inspired by the super liquid transportation property of the asymmetric capillary ratchet of the bird beak and the peristome surface of the pitcher plant. Through combing the asymmetric capillary rachet induced water transportation property and the microcavity array induced water directional transportation property on the biomiemtic 3D evaporator, water film that unidirectionally spreads on the 3D evaporator displays thickness gradient, which leads to enhanced solar-dirven water evaporation and localized salt crystallization. **a** The super liquid transportation property of the asymmetric capillary ratchet of the bird beak. **b** The super liquid transportation property of the peristome surface of the pitcher plant. **c** The inhomogeneous water film induced localized salt crystallization on the biomimetic 3D evaporator and its application in solar-driven water evaporation enhancement. **d** Schematic configuration of size-dependent resin refilling induced additive manufacturing based on the continuous DLP 3D printing system. Inset is the scheme of the size-dependent particle refilling process where particles with a dimension larger than the slicing thickness cannot flow along with the refilling resin and are solidified only on the surface of the printed structure. **e–j** Characterization of the biomimetic 3D evaporator. **e** Side view reconstructed Micro-CT image of the biomimetic 3D evaporator. **f** Top-angled cross-sectional Micro-CT image of the biomimetic 3D evaporator. **g** Side cross-sectional Micro-CT image of the biomimetic 3D evaporator that displays the microcavities with dimensional gradient. **h** Top-angled cross-sectional SEM image of the biomimetic 3D evaporator. **i** Enlarged view of (**h**). **j** Side cross-sectional SEM image that demonstrates the micropores only distribute on the surface of the biomimetic 3D evaporator.

the evaporator surface, while the surface distributed micropores can enhance the suctioned water spreading across the asymmetric grooves. It should be noted that the water upward moving velocity on the biomimetic 3D structure based on the microcavity induced water continuous and inward liquid transportation, is much faster than that on porous filter paper originated from porosity induced capillary wicking (Supplementary Fig. 6). To demonstrate the effective water coverage on the biomimetic structure, the 3D wetting state of the structure is characterized through microcomputed tomography (Micro-CT), with the bottom of the 3D structure immersing into the liquid bath (Fig. 2e) during the X-ray reconstruction process. As shown in Fig. 2f, the liquid is trapped in the microcavities with a thin layer of liquid on the sidewall of the 3D structure. Due to the asymmetric groove and the microcavity dimensional gradient, a 3D water film with thickness inhomogeneity along the sidewall is spontaneously formed, where the apex liquid film (~15 μm the thinnest) is thinner than the bottom liquid film (~1500 μm the thickest), as displayed in Fig. 2g, h.

**Solar steam enhancement**. We further investigate the solar-driven water evaporation performance through floating the evaporator on the water surface. The evaporation rates are examined by recording the mass change under one sun illumination or in darkness. For comparison, a 2D plane (diameter of 11.0 mm and

thickness of 1.0 mm) is prepared from the same composite resin through additive manufacture. Typical curves of time-dependent mass change for pure water, the 2D plane, and the biomimetic 3D evaporator are measured and plotted through a self-made apparatus (Supplementary Fig. 7, U-typed tube). As shown in Fig. 2i, under one sun illumination, the water evaporation rate increases from ~0.41 kg m$^{-2}$ h$^{-1}$ for pure water to ~1.07 kg m$^{-2}$ h$^{-1}$ for the 2D plane, and to ~2.28 kg m$^{-2}$ h$^{-1}$ for the biomimetic 3D evaporator. In darkness, the biomimetic 3D evaporator can also increase water evaporation rate, where evaporation rates are ~0.11 kg m$^{-2}$ h$^{-1}$ for pure water and ~0.84 kg m$^{-2}$ h$^{-1}$ for the biomimetic 3D evaporator (Fig. 2j), while the 2D plane can only slightly increase the water evaporation rate to ~0.20 kg m$^{-2}$ h$^{-1}$. Furthermore, we also prepare a control 3D evaporator with the same 3D morphology but without the addition of photothermal material CNTs, and investigated its water evaporation rate in darkness (Supplementary Fig. 8a). The equivalence of the evaporation rates of the 3D evaporators with and without photothermal materials indicates that the introduction of the evaporator with biomimetic 3D morphology can greatly enhance water evaporation in darkness.

## Discussion
To investigate the mechanism of the solar-driven evaporation enhancement of the biomimetic 3D evaporator, an infrared

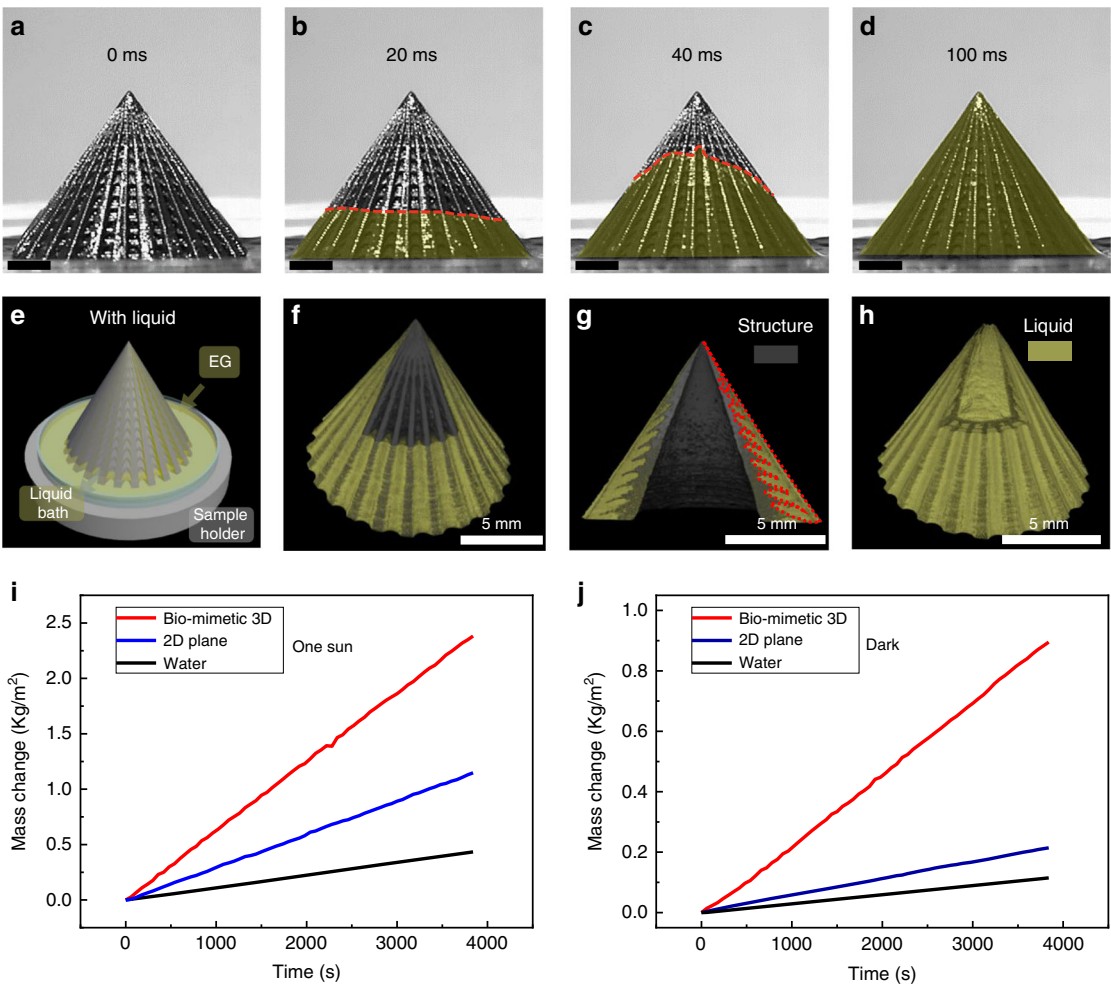

**Fig. 2 Inhomogeneous water film induced solar-driven water evaporation enhancement. a–d** Time sequence of optical captures displaying the ultra-fast water upward spreading process on the biomimetic 3D evaporator surface. Scale bars are 2 mm. **e–h** 3D reconstruction of the equilibrium state of the biomimetic 3D evaporator in the wet state through Micro-CT from different viewing points. Panels (**e**), (**f**), and (**g**) are the scheme of the Micro-CT configuration, side view Micro-CT, and cross-sectional Micro-CT images of the reconstructed wet sample, respectively. Ethylene glycol (EG) with high boiling poit is utilized as the test liquid to reduce the influence of the liquid evaporation during Micro-CT characterization process. The red dot line in (**g**) represents the contact line between liquid and the microcavity structure of the biomimetic 3D evaporator. **h** Micro-CT image of the solely reconstructed hierarchical water pathway with thickness gradient generated on the biomimetic 3D evaporator surface. **i** The mass change of the water on the biomimetic 3D evaporator under one sun illumination (1 kW m$^{-2}$), with pure water and 2D plane as controls. Red, blue, and black lines represent mass-change curves of the biomimetic 3D structure, the 2D plane, and pure water under one sun illumination, respectively. **j** The mass change of the water on the biomimetic 3D evaporator in darkness, with pure water and 2D plane as controls. Red, blue, and black lines represent mass-change curves of the biomimetic 3D evaporator, the 2D plane, and pure water in darkness, respectively.

camera is employed to monitor the temperature evolution of the water film formation process and the dynamic state of the solar steam generation process. Figure 3a reveals the temperature mapping of the biomimetic 3D evaporator in dry state without illumination, where it possesses a homogeneous temperature profile as there is no energy input. Once under illumination, the whole structure can be heated with the increasing of time (Fig. 3b). Temperature inhomogeneity along the dry 3D evaporator sidewall is formed, where the temperature of the apex structure exceeds that of the bottom structure during the whole illumination process. As light is perpendicularly illuminated to the projected area of the 3D structure, the energy that the dry 3D structure can acquire from the illumination differentiates along the sidewall ascribing from the conical morphology. With a closer position from the light source, higher energy intensity can be acquired at the apex. Cooperating with a larger specific surface area at the apex, it will lead to a higher temperature of

the apex structure. In other words, the energy that the structure can absorb is position-related, i.e., the position-related utilization feature of the input energy. The analyses agree well with the experimental data measured by the infrared camera (Fig. 3b), which further proves our explanation on the temperature gradient generation on the dry structure. Therefore, the temperatures at the apex and at the bottom are selected as representatives and are monitored vs. time (Supplementary Fig. 9).

Floating the biomimetic 3D evaporator on a water surface, as Fig. 3c, d reveals, the self-climbed water coverage leads to surface temperature distribution variation. If without considering the evaporation process, the apex liquid film with smaller thickness is much easier to maintain a higher temperature than the thicker liquid film at the bottom. As water evaporation cannot be ignored, a thinner liquid film spreading at the higher temperature apex structure means a higher tendency to

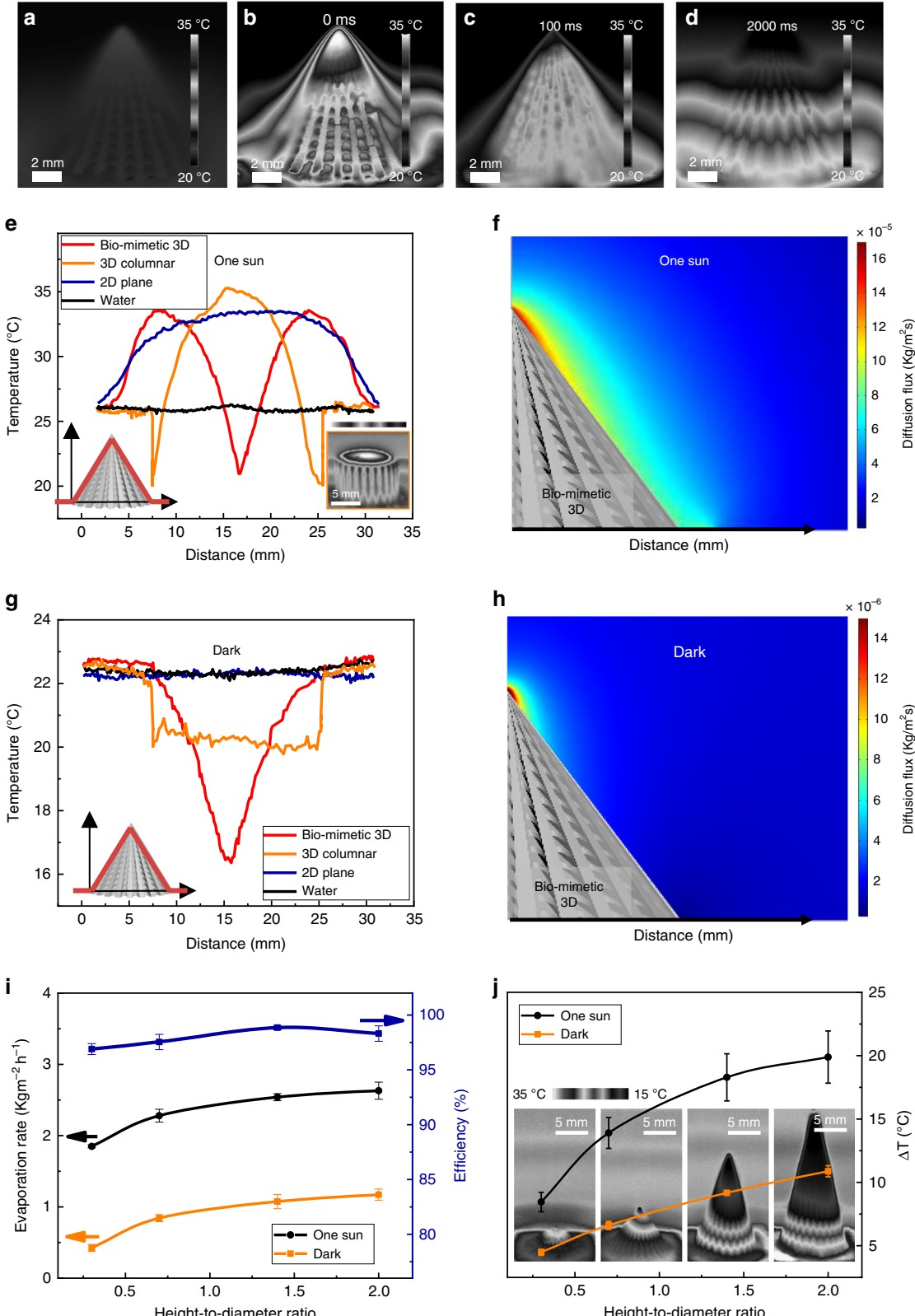

evaporate, indicating the position-related water evaporation phenomenon along the liquid film. Numerical simulation is further employed to prove the existence of the position-related water evaporation phenomenon, the result of which is consistence with the above analysis, where the steam diffusion flux at the apex is larger than that at the bottom under one sun illumination as shown in Fig. 3f. Competition process thus

exists in between the position-related energy absorption and transfer, and the position-related water evaporation. Infrared camera image reflects the final competition result of the surface temperature profile of the evaporator, as shown in the red line in Fig. 3e, where an inversion of the temperature gradient is achieved on the sidewall after covering the liquid film and reaching the equilibrium state. The temperature of the apex

**Fig. 3 Mechanism of the solar steam enhancement on the biomimetic 3D solar evaporator. a** Infrared image of the biomimetic 3D evaporator in dry state in darkness. **b**, **c** Time sequence of infrared captures during the water upward spreading process. **d** Infrared image of the dynamic equilibrium state of the solar steam generation process. **e** Temperature profiles along the biomimetic 3D evaporator surface (red line) under one sun illumination, with 3D columnar structure (orange line), 2D plane (blue line), and pure water (black line) as controls. Inset is the equilibrium infrared image on the 3D columnar structure under one sun illumination. Temperature range is 15–35 °C. **f** Numerical simulation result of the steam diffusion flux at the water/steam interface under one sun illumination. **g** Temperature profiles along the biomimetic 3D structure surface (red line) in darkness, with 3D columnar structure (orange line), 2D plane (blue line), and pure water (black line) as controls. **h** Numerical simulation result of the steam diffusion flux at the water/steam interface in darkness. **i** The mass change and energy efficiency of water on the biomimetic 3D evaporators with different H/D ratios. Orange, black, and blue lines represent the water evaporation rate in darkness, the solar steam generation rate under one sun illumination, and the energy efficiency under one sun illumination, respectively. The error bars in the evaporation rate result from errors in the mass-change measurements. The error bars in the efficiency values resulted from errors in the measurement of solar illumination power, the evaporation rate and the interface temperature. Each error bar represents the deviation from at least five data points. **j** Temperature differences between the bottom and the apex surface temperature with the variation of H/D ratios. Black and orange lines represent the temperature difference under one sun illumination and in darkness, respectively. Insets are equilibrium infrared images on biomimetic 3D evaporators with different H/D ratios under one sun illumination. The error bars resulted from errors in the measurement of the surface temperature. Each error bar represents the deviation from at least five data points.

liquid film (blue line in Supplementary Fig. 9) is much lower than the bottom (black line in Supplementary Fig. 9), indicating that water evaporation dominates which takes heat away and decreases the interfacial temperature at the apex, while the energy absorption and transfer process dominate at the bottom. It is worth mentioning that the thickest bottom liquid film can be heated to ~34.5 °C after reaching the equilibrium state, indicating the sufficient energy transfer from the biomimetic 3D structure to the liquid film, basing on the designed groove structure and the contacting mode of water with corresponding grooves, as displayed in Supplementary Fig. 10.

The temperature gradient also occurs in darkness, just as displayed in the red line of Fig. 3g, which is consistence with the simulation result in Fig. 3h, indicating more vigorous water evaporation and lower surface temperature at the apex. Therefore, the position-related water evaporation feature occurs both in darkness and under one sun illumination but is enhanced under illumination, as shown in Fig. 3f–h. In addition, the wet biomimetic 3D evaporators with and without CNT (Supplementary Fig. 8b) possess almost the same temperature distribution across the entire structure in darkness, further proving the function of the structure and consequent water film inhomogeneity on water evaporation enhancement. Different from the biomimetic 3D evaporators, the temperature profiles of the 2D plane surface (blue lines in Fig. 3e–g) and pure water (black lines in Fig. 3e–g) display homogeneous distribution along the interface both in darkness and under solar illumination (Supplementary Fig. 11). Therefore, the formulation of temperature gradient of the biomimetic 3D evaporator under one sun illumination and in darkness can be attributed to the position-related competition between the energy absorption and transfer inside the system, and the position-related evaporation along the water film. Further combining with the water evaporation rates data in Fig. 2i, j, the enhancement of biomimetic 3D evaporator rests on the generation of structure inhomogeneity. With continuous evaporation, water can be fully evaporated on the evaporator with no water left (Supplementary Movie 2).

The temperature gradient will further induce surface tension difference inside the liquid film, i.e., the well-known Marangoni effect. It will provide a thermocapillary force, which can be expressed as Eq. (1), inside the liquid film to drive the liquid transportation, which can be expressed as Eq. (2), from high temperature to low temperature, i.e., from the bottom liquid film to the apex liquid film. As we have simulated and analyzed that the apex possesses higher steam flux both in darkness and under one sun illumination, water is thus continuously transported to the site with a higher evaporation rate, which can realize effective

and continuous water evaporation. Therefore, the unevenly utilized input energy is further reutilized in the form of thermocapillary force, i.e., the energy reutilization property of this system. The thermocapillary force $\tau$, originated from temperature difference can be expressed as[41,42]

$$\tau = \Delta\gamma/L = \frac{\gamma_{\mathrm{L}} - \gamma_{\mathrm{H}}}{L} = \frac{\mathrm{d}\gamma}{\mathrm{d}T} \cdot \frac{\Delta T}{L} \qquad (1)$$

where $\Delta\gamma$ is surface tension difference, $\gamma_{\mathrm{L}}$ and $\gamma_{\mathrm{H}}$ are the surface tensions of liquid at low temperature and high temperature, respectively. $\mathrm{d}\gamma/\mathrm{d}T$ is the coefficient of surface tension as a function of temperature, $\Delta T$ is the temperature difference between the apex liquid film and the bottom liquid film, $L$ is the distance between the two positions. The liquid transportation inside the liquid film under the function of thermocapillary force can be expressed as[43]:

$$\tau \sim \eta \frac{v_{\mathrm{T}}}{e} \qquad (2)$$

where $v_{\mathrm{T}}$ is the velocity induced by thermocapillary force, $e$ is the effective thickness of the liquid film. Combining Eqs. (1) and (2), the liquid transportation flux $Q_{\mathrm{T}}$ induced by temperature difference can be expressed as

$$Q_{\mathrm{T}} \sim v_{\mathrm{T}} \cdot S \sim \frac{S}{\eta} \cdot \frac{e}{L} \cdot \frac{\mathrm{d}\gamma}{\mathrm{d}T} \cdot \Delta T \qquad (3)$$

where $S$ is the effective cross-sectional area along the water moving direction. Thus, $Q_{\mathrm{T}}$ is in direct relation with $\Delta T$, indicating that enhanced supplement of water to the apex with a higher evaporation rate can be kept during the continuous evaporation process.

For the wet biomimetic 3D evaporator under one sun illumination, the temperature difference between the apex liquid film (~21.0 °C, blue line in Supplementary Fig. 9) and the bottom liquid film (~34.5 °C, black line in Supplementary Fig. 9) is about 13.5 °C, the water upward moving flux $Q_{\mathrm{T}}$ calculated from Eq. (3) is about $2.67 \times 10^{-4}$ g s$^{-1}$, which can meet the need for the timely water evaporation ($1.81 \times 10^{-4}$ g s$^{-1}$). It is worth mentioning that the apex liquid film temperature is ~21.0 °C under one sun illumination, which is ~4.0 °C lower than the ambient temperature. Based on the previous investigations[11,12], energy can also be directly collected from the surrounding environment, which contributed cooperatively with thermocapillary force to enhance solar-driven water evaporation.

Another control structure, a 3D columnar structure, is prepared to prove that the generated thermocapillary force can indeed enhance water evaporation and energy efficiency.

Compared with the biomimetic 3D structure, the 3D columnar structure possesses the same projected area, the same height and the same H/D ratio. The microcavities along each groove possess the same dimension, which results in a liquid film with a homogeneous thickness (Supplementary Fig. 12). In darkness, the surface temperature of the 3D columnar structure is higher than that on the biomimetic 3D structure. However, the water evaporation rate on the 3D columnar structure is lower than that on the biomimetic 3D structure (Supplementary Table 2, orange line in Fig. 3g). Without light, more energy is gathered from the surrounding environment for the biomimetic 3D structure. The contribution of thermocapillary force in enhancing water evaporation in darkness is hard to distinguish for the two structures.

Under one sun illumination, the average surface temperatures of both the 3D columnar structure and the biomimetic 3D structure are almost the same, indicating that the energy capable from the surrounding environment can be considered as the same (Supplementary Table 2). However, the temperature profiles on the surfaces of the two structures are different. On the 3D columnar structure, the top surface temperature is higher than the bottom (orange line, Fig. 3e), whose trend is contrary to the biomimetic 3D structure, which leads to the opposite direction of the thermocapillary forces. The direction of the water flow induced by thermocapillary force for the 3D columnar structure is thus from top to bottom, where the water supplementation from the source to the evaporation surface is hindered during the continuous evaporation process. Without sufficient water supplementation, the water used for solar-driven evaporation is thus decreased, which finally leads to the reduced water evaporation rate comparing with the biomimetic 3D structure. In our system, the thermocapillary force can carry water from the bottom to the apex, the more vigorous evaporation site, to realize effective water evaporation and enhance energy efficiency.

The above conclusion is established on the default condition that the water supplementation speed can match the water evaporation speed, which maintains a continuous liquid film for water evaporation. If the water supplementation speed cannot satisfy the water evaporation speed, the temperature gradient will be inversed as shown in Supplementary Fig. 13a, b on the smooth conical structure at first, where the temperature of the apex liquid film is higher than the bottom liquid film. However, the apex liquid film will be completely evaporated which finally leads to the dewetting of the liquid film on the smooth conical structure and the decrease of the effective contacting surface of water/structure. The water evaporation rate on such structure is much lower than that on the biomimetic 3D structure (Supplementary Fig. 13c). Therefore, the effective water supplementation on the biomimetic 3D structure also contributes to the realization of effective water evaporation.

We further investigate the influence of the 3D evaporator morphology on the enhancement of solar-driven water evaporation. Another three evaporators with different H/D ratios (including one shorter and another two taller than previously used, with H/D ratios of 0.3, 1.4, and 2.0, respectively) are prepared through regulating the number of microcavities along each asymmetric groove without changing the initial dimension of the microcavity, as shown in Supplementary Fig. 2b, c. As indicated in the orange line of Fig. 3i, with the structure H/D ratio increasing from 0.3 to 0.7, 1.4, and 2.0, respectively, the solar-driven water evaporation rate increases from ~1.85 to ~2.28, ~2.54, and ~2.63 kg m$^{-2}$ h$^{-1}$, respectively. As displayed in Supplementary Fig. 14a, the bottom temperature has little change with the increasing of H/D ratio, while the apex liquid film temperature decreases drastically. Thus, the temperature difference increases (Fig. 3j), which leads to larger and larger

thermocapillary force inside the liquid film with the increasing of H/D ratio. In addition, the energy can be acquired from the surrounding environment increases owing to the increased temperature difference between the apex liquid film and the surrounding environment, which further leads to the increasing trend of solar-driven water evaporation rate. The same tendency is acquired in darkness, as shown in Supplementary Fig. 14b and black lines in Fig. 3i, j, which further explains the increased water evaporation rate on the 3D evaporator in darkness as displayed in Fig. 2j. If the water evaporation rate in darkness is deduced from the water evaporation rate under one sun illumination, the net evaporation rates are ~1.43, ~1.44, ~1.47, and ~1.46 kg m$^{-2}$ h$^{-1}$ for evaporators with H/D ratio of 0.3, 0.7, 1.4, and 2.0, respectively. The net evaporation rates of the biomimetic 3D evaporators remain as a constant value, which approaches the theoretical water evaporation rate.

The energy efficiency ($\eta$) of the biomimetic 3D evaporator is calculated as the percentage of the energy that is utilized by net water evaporation compared with the total energy of the incident sunlight to evaluate the photothermal performance. It is generally calculated via the following formula[11,44]:

$$\eta = m(L_v + Q)/P_{in} \tag{4}$$

Where $m$ is the net water mass-change rate (kg m$^{-2}$ h$^{-1}$), $L_v$ is the latent heat of water evaporation (J kg$^{-1}$)[45], $Q$ is the sensible heat of water (J kg$^{-1}$), and $P_{in}$ is the power of the incident simulated sunlight beam (J m$^{-2}$ h$^{-1}$). Due to the temperature gradient on the 3D evaporator, the energy efficiencies of different H/D ratios are calculated based on the average water/structure temperatures calculated from Supplementary Fig. 14a. The average interfacial temperature decreases from ~30.9 °C to ~28.9, ~22.7, and ~21.0 °C with the increasing of H/D ratio from 0.3 to 0.7, 1.4, and 2.0, respectively. The energy efficiencies calculated from the net evaporation rates are ~96.9, ~97.5, ~98.9, and ~98.3% correspondingly (blue line in Fig. 3i). In the control experiment, the 2D plane has an average interfacial temperature of ~33.1 °C and the net evaporation rate of ~0.87 kg m$^{-2}$ h$^{-1}$, the energy efficiency of which is only ~59.2%. Comparing with other structures, the biomimetic 3D evaporator has the highest efficiency, which can be attributed to the reutilizing of solar energy through the formation of temperature gradient that allows for additional energy capture from the ambient environment.

High salinity, 25.0 wt% of sodium chloride (NaCl) solution, is prepared as the representative brine sample to demonstrate the solar desalination capability and durability of the biomimetic 3D evaporator. The desalination process is recorded by a camera. With continuous solar illumination and water evaporation, NaCl will crystallize on the evaporator as the concentration is approaching the critical concentration of the saturated solution (~26.4 wt%, 25 °C). The salt crystallization or accumulation position was spatially located at the apex of the 3D evaporator (Fig. 4a–d). As position-related water evaporation occurs along the liquid film (Fig. 3f–h), cooperating with the liquid film thickness gradient, a salinity gradient further generates along the sidewall of the biomimetic 3D evaporator, where the salt concentration of the apex liquid film is higher than that of the bottom liquid film. The apex liquid film is easier to reach the critical crystallization concentration compared with the bottom during the continuous water evaporation process, i.e., the higher the position on the biomimetic 3D evaporator, the easier for NaCl crystallization. For such a high salinity, NaCl will also crystallize on the sidewall of the 3D evaporator surface. Whereas, the crystallized NaCl on the sidewall can also flow along the spontaneously formed water flow from the bulk water and move upwardly to the apex of the biomimetic 3D

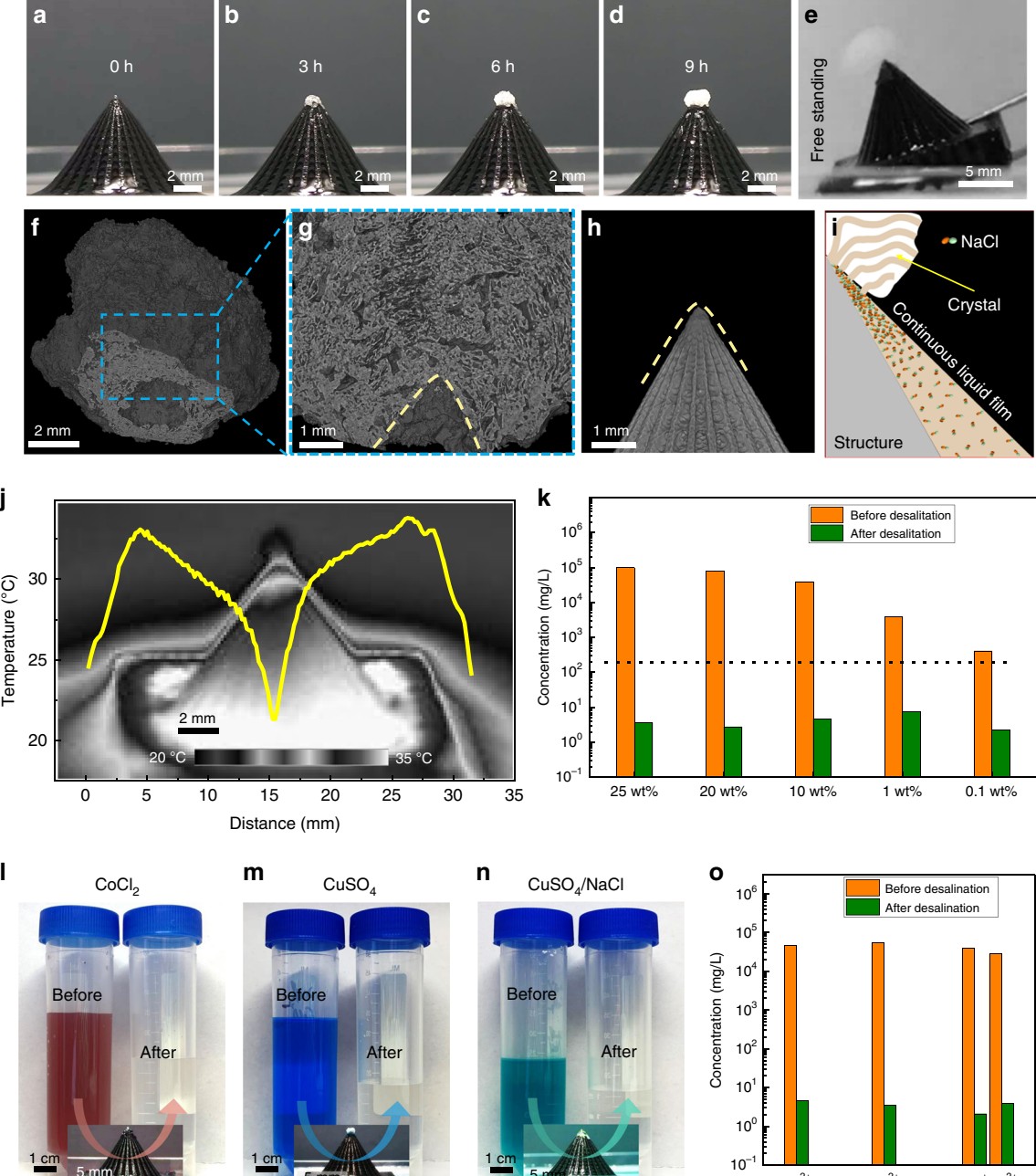

**Fig. 4 Localized salt crystallization mechanism on the biomimetic 3D evaporator. a–d** Time sequence of optical captures displaying the localized crystallization process on the biomimetic 3D evaporator. **e** Optical image of the salt easy-removing property of the biomimetic 3D evaporator, the crystallized salt can be removed through leaning the evaporator. **f** Bottom view Micro-CT image of the removed NaCl crystal. **g** Side cross-sectional Micro-CT image of the detached crystal, which amplifies the conical hole morphology of the contacting interface. **h** Micro-CT image of the apex position of the biomimetic 3D evaporator. **i** Scheme of the continuous water film along the sidewall of the biomimetic 3D evaporator which extends to the localized crystal at the apex position. **j** Temperature profile along the sidewall of the biomimetic 3D evaporator with the existence of the localized crystal at the apex position. Inset is the corresponding infrared image. **k** Measured concentrations of $Na^+$ in different brine samples (including 25, 20, 10, 1, and 0.1 wt% NaCl solution) before and after desalination. Orange and green columns represent the metal ion concentrations before and after purification, respectively. The broken line refers to the WHO $Na^+$ concentration standards for drinkable water. **l–n** Optical image of the water sample with different heavy metal ions (including the aqueous solution of 10 wt% $CoCl_2$, 20 wt% $CuSO_4$, and mixed solution composed of 10 wt% NaCl and 10 wt% $CuSO_4$) before and after solar purification. **o** Measured concentrations of corresponding metal ions before and after purification. Orange and green columns represent the metal ion concentrations before and after purification, respectively.

evaporator (Supplementary Fig. 15, Supplementary Movie 3). Significantly, the salt free stands on the 3D evaporator and can be easily removed through leaning the evaporator (Fig. 4e, Supplementary Movie 4). After characterizing the detached salt

through Micro-CT, a 3D salt/evaporator contacting surface with conical hole morphology inside the salt is clarified (Fig. 4f). As shown in Fig. 4g, h, the dimension of the conical hole is larger than the apex of the biomimetic 3D evaporator, which means

that the crystallized salt is not directionally contacting with the 3D evaporator, but with a liquid layer in between (Fig. 4i). Even with salt crystallization at the apex, the solar steam generation rate does not obviously change ($\sim$2.24 kg m$^{-2}$ h$^{-1}$ under one sun illumination, and $\sim$0.81 kg m$^{-2}$ h$^{-1}$ in darkness). Micro-CT images show that channels can be found inside the salt (Fig. 4g and Supplementary Movie 5). The continuous water film pathway was thus extended inside the crystallized salt at the apex (Fig. 4i). In addition, the temperature profile (Fig. 4j) during the solar desalination process shows the same tendency and value with that using pure water (red line, Fig. 3e). With the average interface temperature of $\sim$30.1 °C and net evaporation rate of $\sim$1.43 kg m$^{-2}$ h$^{-1}$, the efficiency under high salinity calculated from Eq. (4) is $\sim$97.1%. Moreover, the localized crystallization phenomenon is universal for evaporators with different H/D ratios (Supplementary Fig. 16).

As shown in Fig. 4k, after desalination, the Na$^+$ concentration characterized by inductively coupled plasma mass spectroscopy (ICP-MS) is significantly decreased by four orders of magnitude to 3.6 mg L$^{-1}$, and is approximately two orders of magnitude below the drinking water standards defined by the World Health Organization (WHO)[46]. Furthermore, it is versatile to brine samples with a lower NaCl concentration of 20.0, 10.0, 1.0, and 0.1 wt% (Fig. 4k). Moreover, the biomimetic 3D evaporator can also purify water with a high concentration of heavy metals. As shown in Fig. 4l–n, the colors of 10 wt% cobalt dichloride (CoCl$_2$) and 20 wt% copper sulfate (CuSO$_4$·5H$_2$O) change from red and blue to clear after solar-driven water evaporation, corresponding to the reduction of Co$^{2+}$ and Cu$^{2+}$ from 45999.2 mg L$^{-1}$ and 53002.5 mg L$^{-1}$ to 3.9 mg L$^{-1}$ and 2.3 mg L$^{-1}$ (Fig. 4o), respectively. For mixed solution composed of 10 wt% NaCl and 10 wt% CuSO$_4$·5H$_2$O, the concentrations of Na$^+$ and Cu$^{2+}$ can be decreased simultaneously, with Na$^+$ and Cu$^{2+}$ decreased from 39447.1 mg L$^{-1}$ and 27787.6 mg L$^{-1}$ to 3.5 mg L$^{-1}$ and 3.7 mg L$^{-1}$, respectively, demonstrating its potential for purifying complex solutions.

A batch purification apparatus composed of an inlet tube, condenser, brine sample container and an outlet tube is set up to simulate the practical water purification process (Fig. 5a, Supplementary Fig. 17)[47]. Evaporator composed of 3D structure array with H/D ratio of 1.4 is prepared, where the intervals of adjacent 3D structures are filled with flat planes (Fig. 5b). Natural seawater sample from Jiaozhou Bay, the Yellow Sea is used as the desalination sample. As shown in Fig. 5c–e and Supplementary Movie 6, the generated vapor continuously condensed on the top and side inner surfaces of the condenser, and then flow to the bottom surface, which is finally collected by the outlet tube. The gradually decreased seawater can be supplemented by the inlet tube to the container, and the purified water can be collected through the outlet tube, which endows the apparatus with continuous water purification property. The water collecting velocity ($\sim$1.72 kg m$^{-2}$ h$^{-1}$) is lower than the open system ($\sim$2.54 kg m$^{-2}$ h$^{-1}$), which can be attributed to the increased humidity in the closed system and residual on the inner wall of the condenser. In addition, the concentrations of all four primary ions (Na$^+$, Mg$^{2+}$, K$^+$, and Ca$^{2+}$) initially present in the seawater sample are significantly reduced (Fig. 5f), indicating effective purification of natural seawater. Solar endurance test in Fig. 5g shows that it presented a stable evaporation rate to the seawater for 10 days under one sun illumination for 9 h every day. The accumulated salt on the apex position of the 3D evaporators can be easily removed and collected (Fig. 5h–j), indicating that the biomimetic 3D evaporator is reliable for long-term solar desalination.

In summary, inspired by the unique water transport property of the asymmetric capillary ratchet of bird beak and the pitcher plant peristome surface, we have constructed a biomimetic 3D evaporator for high-efficiency solar-driven water evaporation and desalination. Based on the developed resin system, the size-dependent resin refilling phenomenon occurs during the continuous additive manufacturing process. Surface distributed micropores are formed on the prepared surface, endowing the biomimetic 3D evaporator with ultra-fast water spreading property. Due to the designed morphology of the 3D structure with asymmetric grooves and the gradient microcavity arrays, the liquid film spreads on the structure surface displays position-related liquid film thickness and temperature gradient along the sidewall, which further leads to the thermocapillary force inside the liquid film and the capability to capture energy from the surrounding environment to enhance water evaporation and energy efficiency. High water evaporation rate of 1.17 kg m$^{-2}$ h$^{-1}$, high solar-driven water evaporation rate of 2.63 kg m$^{-2}$ h$^{-1}$ and high energy efficiency of larger than 96% can be achieved under one sun illumination with excellent stability even under high salinity (25 wt% NaCl solution). The locally crystallized salt free stands at the apex without contaminating the evaporator, and can be easily removed due to the extension of the inhomogeneous water film inside the crystal. In addition, energy efficiency and water evaporation are not influenced by salt accumulation, indicating its potential for sustainable and practical applications in the future.

## Methods

**Preparation of the nanocomposite.** The UV curable resin system was formulated by mixing prepolymer, reactive dilute, photo initiator and other additions which all tailored to be active at the relevant wavelength of UV. Here, polyacrylate system composed of prepolymer acrylic resin, monomer di(ethylene glycol) ethylether-acrylate, photo initiator 2,4,6-phenylbis(2,4,6-trimethylbenzoyl)phosphine oxide, and crosslinker poly(ethylene glycol) diacrylate 700 was employed. Carbon nanotubes (CNTs) with a mean diameter of 100 nm, a length range of 20–200 μm were purchased from Sigma-Aldrich. Citrate sodium was an analytical grade and bought from Beijing Chemical Works (China). The individual components of the UV curable resin were mixed before the dispersion process. Then the carbon nanotube powder and the citrate sodium powder were mixed with the UV curable resin separately to form two mixtures. Then, the two mixtures were further mixed to form a slurry. The weight percentages of CNTs and citrate sodium in the slurry were 0.5 wt% and 20 wt%, respectively. The slurry was transformed into the ultrasonic cell pulverizer to achieve a stabilized dispersion. Finally, after degassing through vacuum the composite resin was acquired.

**Post-3D printing treatment.** After 3D printing, printed parts were developed in ethanol for 2 min with ultrasonic treatment to remove the uncured resin, then developed in a 1:1 vol/vol solution of ethanol and water to remove the citrate sodium solidified on the surface. To enhance the mechanical properties, a post-curing process was performed in a tank with 20 multidirectional LEDs emitting 405 nm light for 30 min at room temperature. Before use, the front side of 3D printed structures was plasma treated to endow the upper surface with hydrophilicity for water spreading. After plasma treatment, surface hydrophilicity increases due to the increased amount of surface –OH. It can be clarified from the X-ray photoelectron spectroscopy (XPS) results as displayed in Supplementary Fig. 18, where the oxygen-containing groups rises. Originating from the bond scission and incorporation of oxygen onto the cured resin surface, surface hydrophilicity is improved after plasma treatment. The detailed characterization strategies are shown in Supplementary Note 2.

**Numerical simulation.** Numerical simulation was performed using a commercial finite element software package COMSOL Multiphysics 5.4. The water evaporation process of the whole system is simulated by solving the below equations:

$$-D\nabla^2 c_v = 0 \qquad (5)$$

$$c_v = \phi c_{sat} \qquad (6)$$

$$g = -D\nabla c_v \qquad (7)$$

where $C_v$ is the concentration of the vapor in the air, $D$ is the diffusion coefficient, $C_{sat}$ is the saturated vapor concentration, $\phi$ is the relative humidity, $g$ is the steam diffusion flux. The vapor concentration at boundaries of the water surface is set to be saturated vapor concentration, $C_{sat}$. The relative humidity of the environment is set as 0.5. The heat transfer is simulated by solving Eq. (8) and Eq. (9):

$$-k\nabla^2 T = Q \qquad (8)$$

where $T$ is temperature, $k$ is heat conductivity coefficient, $Q$ is a heat source. The heat

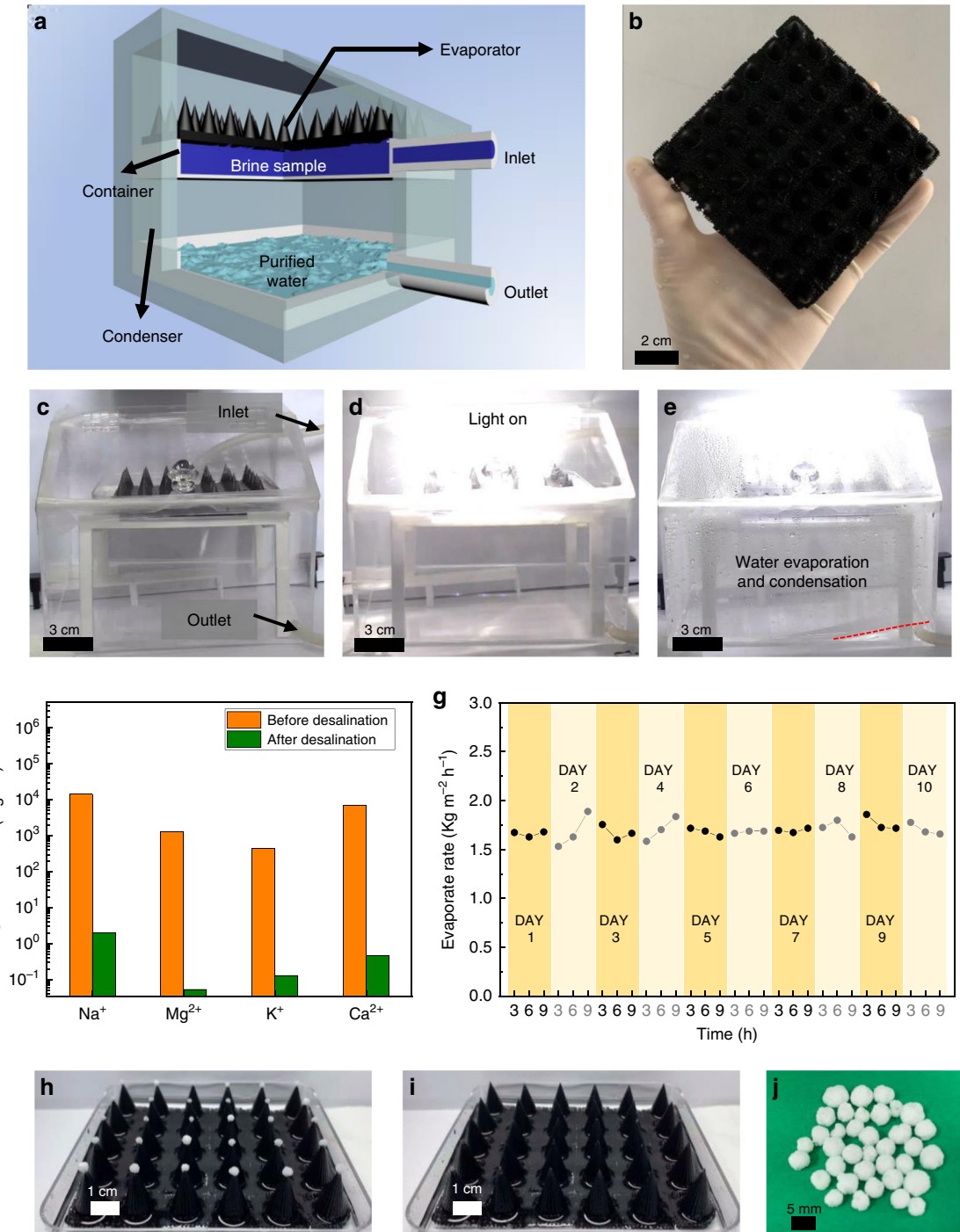

**Fig. 5 Solar desalination and durability of the biomimetic 3D evaporator. a** Scheme of the batch purification prototype which simulates the practical solar water purification apparatus. Brine sample is introduced into the container through the inlet. Water evaporates on the 3D evaporator surface under illumination, and then condenses on the upper surface and sidewall of the condenser, which is finally collected by the outlet. **b** Optical image of the large area evaporator composed of arrays of biomimetic 3D structures with H/D ratio of 1.4. **c** Optical image of the batch purification prototype without solar illumination. **d** Optical image of the batch purification prototype under solar illumination, where most light is collected by the evaporator. **e** Optical capture of the generated solar steam condensing process on the inner wall of the condenser, which is finally collected by the outlet tube. Red dashed line represents the upper surface of purified water before being output by the outlet. **f** Measured concentrations of four primary ions in the actual seawater sample from Jiaozhou Bay, the Yellow Sea before and after desalination. Orange and green columns represent the metal ion concentrations before and after purification, respectively. **g** Solar endurance test results of arrayed biomimetic 3D evaporator continuously exposed under one sun illumination in the closed system batch purification prototype for 10 days with 9 h every day. **h, i** Optical images of the arrayed evaporators before (**h**) and after (**i**) removing the locally crystallized salt. **j** Optical image of the collected salt detached from the arrayed evaporator after the solar endurance test.

source at the boundaries of water surfaces is set to be $Q_{evap}$ as shown below:

$$Q_{evap} = -L_v g_{evap} \qquad (9)$$

Where $g_{evap}$ is the diffusion flux at the water surface, $L_v$ is coefficient latent heat. Another heat source is set at the boundaries of solid surfaces to represent the heat from illumination. The environment temperature is set as 293.15 K. $C_{sat}$ increases with temperature $T$, which is taken into account in the simulation. All the parameters of $D$, $k$, $C_{sat}$, $L_v$ are taken from build in material library in COMSOL Multiphysics 5.4. Because of the symmetry of the structure, one groove of the wet biomimetic structure in darkness and under one sun illumination is simulated. The detailed geometry of the model used for numerical simulation, and detailed boundary conditions are shown in Supplementary Figs. 19, 20 and Supplementary Table 3.

## Data availability

The authors decalre that the main data supporting the findings of this study are contained within the paper. All other relevant data are available from the corresponding author upon reasonable request.

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

## Acknowledgements

We acknowledge funding of the National Key R&D Program of China (Grant Nos. 2018YFA0208501, 2016YFB0401603, 2016YFC1100502, and 2016YFB0401100), the National Natural Science Foundation (Grant Nos. 51803219, 51773206, and 91963212), K.C. Wong Education Foundation and Beijing National Laboratory for Molecular Sciences (BNLMS-CXXM-202005). N.X.F. and T.G. are grateful for the seed provided by the MIT Energy Initiative.

## Author contributions

L.W. and Z.D. contributed equally to this work. Y.S. conceived and designed the experiments. L.W., Z.D., C.L., C.Y. and Y.Z. performed the experiments. Y.S., L.W., and Z.D. analyzed the data. Z.C., T.G., and N.F. conducted the numerical simulation. L.W. and Z.D. wrote the original paper, Y.S. helped revise it. All authors discussed the results and commented on the paper.

## Competing interests

The authors declare no competing interests.

**Additional information**

