## [Peer Review File · Nature Communications]

Reviewers' comments:

Reviewer #1 (Remarks to the Author):

The authors are well aware of the current problems in desalination using the solar steam generation, and presented creative solutions using knowledge gained from nature. Through a desalination study using a cone-shaped structure with super liquid transportation property obtained from nature, they have newly discovered how the temperature difference formed through evaporation implements the salt spatialized crystallization feature. I think this research is a truly influential and substantial achievement that can accelerate the practical use of solar-steam desalination. Therefore, I recommend publication of this manuscript in Nature Comm. I only have minor comments and questions.;

1. providing effective water/steam transport interface.
2. Spatialized salt crystallization feature > localized salt crystallization feature.
3. DLP > explain full name once in the manuscript.
4. Authors mentioned "It should be noted that the water upward moving velocity on the 3D structure is much larger than that on porous filter paper (Figure S5)." This means probably faster not larger. By the way, Figure S5 c displays wrong indication for filter paper (red) and mimic (black).
5. Is fast upward moving flow always the best for the highest evaporation efficiency? Water film can also reflect light.
6. for the bio-mimetic 3D evaporator (Figure 2j). While, the 2D plane can only slightly increase > for the bio-mimetic 3D evaporator (Figure 2j), while the 2D plane can only slightly increase
7. higher structure temperature > need other expression
8. "Therefore, the input solar energy will generate temperature gradient to enhance water evaporation, while the generated gradient will be reused in the form of thermocapillary force to further enhance water evaporation, which leads to the energy reutilizing property of the bio-mimetic 3D evaporator." Please clarify the exact concept of energy reutilizing property somewhere before this sentence.
9. "Such high efficiency can be attributed to the reutilizing of solar energy through temperature gradient and the additional energy capture from the ambient environment." Based upon my understandings, the sentence means that "Such high efficiency can be attributed to the reutilizing of solar energy through the formation of temperature gradient which allows for additional energy capture from the ambient environment."
10. Additionally, the energy can be acquired from the surrounding environment increases owing to the decreased apex temperature environment increases?

11. When authors calculated the evaporation rate, what unit area did authors use? I mean projected area or real cone surface area? Probably, projected area, I guess. Why does the net evaporation rate (ie, energy efficiency) decrease slightly when the H / D ratio increases from 1.4 to 2.0?

12. Did authors consider the light modulation effect with respect to height especially during measuring the evaporation rate of high H/D ratio cone?

13. The authors suggest a high energy efficiency of over 96%. To better highlight the reutilizing solar energy effect of the devices developed by the authors, it would be better to present and compare the energy efficiency of the 2D plane.

14. "Before use, the front side of 3D printed structures were plasma treated to endow the upper surface with hydrophilicity for water spreading." ☐ For better understandings of hydrophilicity, please supply XPS result for upper surface in supporting information.

Reviewer #2 (Remarks to the Author):

In this study, the authors reported a novel 3D design for solar steam generation system which achieved high water evaporation rate and spatialized crystallization from high salinity water. At first glance, the solar water vaporizing performance is indeed excellent, especially for high salinity solutions. However, some of the explanations are not fully supported by solid evidences. There are still some other issues need to be studied in-depth. Considering the novelty and contributions to the field, I think the paper could be published in NC after these concerns are addressed. Below are the specific comments.

1. The introduction needs to be improved. Particularly more details need to be added. The literature review failed in covering the state-of-art of studies in the solar steam field, which is critical for readers to understand the research gap this study aimed to address.

2. In the first part of the discussion, the authors tried to illustrate the energy reutilisation through thermocapillary force. As described, the thermocapillary force can drive the liquid transportation from high temperature to low temperature. Nevertheless, how the thermocapillary force enhanced energy reutilisation and water evaporation? Moreover, the explanation (line 18-22, page 6) on the formation of temperature distribution under 1 sun or in the dark is not convincing. Detailed calculation or simulation may be helpful.

3. In this paper, the enhanced water evaporation performance mainly due to the efficient dark evaporation, which was reported $0.84 \sim 1.17 \text{ kg m}^{-2} \text{ h}^{-1}$ for the 3D evaporator. This enhancement can be well explained by the energy harvesting from the environment (Joule, 2018, 2, 1331-1338). So how to prove and demonstrate the real benefits of thermocapillary force? More experimental results and explanations are required.

4. The spatialized salt crystallization is not well explained. The reason why crystallization happened at the apex (Line 16-18 on page 10) might not be because of its faster evaporation. In addition, the statement “the inhomogeneous temperature distribution also leads to spatialized salt crystallization feature on the 3D evaporator” (line 12, page 3) is not well proved in the results and discussions. Actually, similar results have been reported (Environmental Science & Technology, 2018, 11822-11830; Energy & Environmental Science, 2019, 12, 1840-1847). Suggest the authors read these papers and improve their discussions on spatialized salt crystallization.

Reviewer #3 (Remarks to the Author):

In this paper, the authors report 3D printing interfacial solar evaporation structure. The structure achieved high evaporation efficiency. The authors invoked Marangoni effect to explain the high efficiency. Overall, the results are interesting. However, I do not feel the paper contains enough new science or technology advances that warrant its acceptance in Nature Communication. My reasons and comments are detailed below.

1. There are many papers published on this topic, and quite high efficiency has been reported. Some at comparable level.

2. Although the reported structure is clearly 3D, it is difficult to say other porous structures are not 3D. In fact, these structures achieved high efficiency because of the large surface area.

3. Comparable efficiency could be reached using much simpler and cheaper structures. It is unlikely such 3D fabricated structures are cost competitive for applications.

The above are nontechnical but support my lack of enthusiasm for the paper. Technical comments are listed below.

1. The claimed high efficiency does not seem to commensurate with the optical measurements. It seems that the absorptance is around 90%. The highest reported efficiency is over 98%. If one calculates the efficiency based on absorbed heat, it would be higher than 100%. This is not explained.

2. Heat transfer mechanisms during the continuous evaporation process and how the illumination of the sunlight boosts the evaporation are not clear to me. At the wavelengths around 10 micron, the penetration length of water is around 20 micron. From Fig.4e, it seems that the water film thickness near the apex is on the order of 100 micron. Thus, the temperature distribution measured by the infrared camera is the average temperature distribution near the liquid-vapor interface. The question I have is what the temperature distribution of the wall temperature of the conical structure is under one sun illumination. My understanding is that the wall temperature at the apex is higher than that at the bottom. Since the water is transparent to the sunlight, the conical structure is a wet fin with varying cross-section under the uniform heat flux. The water film thickness is thinner near the apex, which is equivalent to the higher heat transfer coefficient near the apex, which leads to the lower liquid-vapor interface temperature near the apex than that at the bottom under one sun illumination. However, it seems that the higher wall temperature at the apex due to the sunlight and the larger heat transfer coefficient near the apex due to a thinner liquid film are competing each other in creating the inverted temperature distribution of the liquid-vapor interface. In other words, if the increasing heat transfer coefficient towards the apex is overridden by the increasing wall temperature towards the apex, the temperature distribution of the liquid-vapor interface may be inverted (higher temperature at the apex). I would like the authors to more clearly explain how the incident sunlight on the surface of the conical structure boosts the evaporation rate. If the conical part is shielded and if the sunlight is incident only on the water bath around the conical structure, does that boost the evaporation further since you create a greater temperature gradient between the bottom and the apex?

3. The authors should explain the side of the container used to test the device. The tested sample is small, and there could be lots of parasitic evaporation from sides. They need to present details of experiments and how various parasitics are considered.

Responses to Reviewer # 1

The authors are well aware of the current problems in desalination using the solar steam generation, and presented creative solutions using knowledge gained from nature. Through a desalination study using a cone-shaped structure with super liquid transportation property obtained from nature, they have newly discovered how the temperature difference formed through evaporation implements the salt spatialized crystallization feature. I think this research is a truly influential and substantial achievement that can accelerate the practical use of solar-steam desalination. that can accelerate the practical use of solar-steam desalination. Therefore, I recommend publication of this manuscript in *Nature Comm*. I only have minor comments and questions.

Reply: We greatly appreciate the reviewer for the positive assessment and valuable suggestions. According to the reviewer's comments, the manuscript has been carefully revised. We hope that the revised manuscript would be suitable for publication in *Nature Communications*.

1. providing effective water/steam transport interface.

Reply: We thank the reviewer very much for the comment. According to the reviewer's suggestion, we have added the scheme of the stream/water transport interface in **Figure 1am** in the revised manuscript. Additionally, we have simulated the solar steam generation process where the profile of the steam diffusion flux (**Figure 3f and Figure 3h** in the revised manuscript) in darkness and under one sun illumination is outlined, which can to some extent represent the water/steam interface.

2. Spatialized salt crystallization feature > localized salt crystallization feature.

Reply: Thanks for the reviewer's suggestion. We have revised the salt crystallization phenomenon from "spatialized" to "localized" in the revised manuscript.

3. DLP > explain full name once in the manuscript.

Reply: The full name of DLP, Digital Light Processing, has been added in Line 22, Page 4 in the revised manuscript.

4. Authors mentioned "It should be noted that the water upward moving velocity on the 3D structure is much larger than that on porous filter paper (Figure S5)." This means probably faster not larger. By the way, Figure S5 c displays wrong indication for filter paper (red) and mimic (black).

Reply: Thanks for the reviewer's comment. The sentence "It should be noted that the water upward moving velocity on the 3D structure is much larger than that on porous filter paper (Figure S5)" has been revised to "It should be noted that the water upward moving velocity on the bio-mimetic 3D structure is much faster than that on porous filter paper (Figure S6)" (Line 3, Page 6 in the revised manuscript). Additionally, we have revised the indications for filter paper and the bio-mimetic

structure in corresponding figure, as shown in **Figure R1** (**Figure S6c** in the revised supplementary information).

Figure R1 (**Figure S6c** in the revised supplementary information). Distance of the water precursor moving on the bio-mimetic structure and filter paper along with time. Black and red lines represent the water precursor moving distance on the filter paper and the bio-mimetic structure, respectively.

5. Is fast upward moving flow always the best for the highest evaporation efficiency? Water film can also reflect light.

Reply: Thanks for the reviewer’s question. To achieve higher evaporation rate and energy efficiency, the supplementation of water at the evaporation interface should be in time. Without sufficient water supplementation, the water evaporation will decrease the amount of sufficient liquid on the evaporator surface for further evaporation, resulting in a discontinuous or limited evaporation process. For example, we have prepared a smooth conical structure (**Figure R2a**, Figure S13a in the revised supplementary information), on which the water supplementation speed cannot satisfy the water evaporation speed. The temperature gradient on the smooth conical structure surface is contrary to the bio-mimetic 3D structure at first (**Figure R2b**, Figure S13b in the revised supplementary information). With continuous illumination, water film will dewet on the smooth conical structure surface and decrease the effective contacting surface of water/structure, which leads to the decreased solar steam generation rate (**Figure R2c**, Figure S13c in the revised supplementary information). Therefore, the effective water supplementation of the bio-mimetic 3D structure also contributes to the realization of effective water evaporation.

Figure R2 (**Figure S13** in the revised supplementary information). **a**. Scheme of the smooth conical structure. The structure possesses the same dimension with the bio-mimetic 3D structure with the only difference in the sidewall

morphology. **b.** Temperature profiles of bio-mimetic 3D evaporator (red line) and the smooth conical structure (green line) under one sun illumination. Insets are corresponding infrared images showing the temperature distribution at the equilibrium state on the bio-mimetic 3D structure and the smooth conical structure under one sun illumination. **c.** The mass change of water on the bio-mimetic 3D structure (red line) and the smooth conical structure (green line) under one sun illumination. The smooth conical structure is prepared from the size-dependent resin refilling induced 3D printing through employing the composite resin in Figure S3.

In our system, the morphology of the water film that fully spreads on the bio-mimetic 3D evaporator is determined by the geometric parameters of the structure. After reaching the stable water film morphology that the 3D structure can afford, water cannot further upwardly move on the 3D evaporator. Besides, during the desalination process in our system, the effective upward supplementation of water on the bio-mimetic 3D structure can form a continuous liquid film. This liquid film can both inhibit the salt crystallization on the solid structure and serve as a lubricant layer for crystallized salt moving to the apex position. Therefore, from these points of view, the water upward moving speed should be as fast as possible.

Corresponding descriptions are added in the revised manuscript (Line 7, Page 12) as: “The above conclusion is established on the default condition that the water supplementation speed can match the water evaporation speed, which maintains a continuous liquid film for water evaporation. If the water supplementation speed cannot satisfy the water evaporation speed, the temperature gradient will be inversed as shown in **Figure S13a-b** on the smooth conical structure at first, where the temperature of the apex liquid film is higher than the bottom liquid film. However, the apex liquid film will be completely evaporated which finally leads to the dewetting of the liquid film on the smooth conical structure and the decrease of the effective contacting surface of water/structure. The water evaporation rate on such structure is much lower than that on the bio-mimetic 3D structure (**Figure S13c**). Therefore, the effective water supplementation on the bio-mimetic 3D structure also contributes to the realization of effective water evaporation”.

6. for the bio-mimetic 3D evaporator (Figure 2j). While, the 2D plane can only slightly increase > for the bio-mimetic 3D evaporator (Figure 2j), while the 2D plane can only slightly increase

Reply: Thanks for the reviewer’s correction. The sentence “for the bio-mimetic 3D evaporator (Figure 2j). While, the 2D plane can only slightly increase” has been corrected to “for the bio-mimetic 3D evaporator (Figure 2j), while the 2D plane can only slightly increase” in the revised manuscript (Line 22, Page 6).

7. higher structure temperature > need other expression

Reply: Thanks for the reviewer’s suggestion. The description of “higher structure temperature” concerns the generation of the temperature gradient of the dry 3D structure and water filled wet 3D structure under one sun illumination. In order to make the temperature gradient generation mechanism clearer, we have revised the “Energy reutilizing through thermocapillary force” section in the **Discussion** part with new descriptions and the addition of new simulation results.

8. **“Therefore, the input solar energy will generate temperature gradient to enhance water evaporation, while the generated gradient will be reused in the form of thermocapillary force to further enhance water evaporation, which leads to the energy reutilizing property of the bio-mimetic 3D evaporator.” Please clarify the exact concept of energy reutilizing property somewhere before this sentence.**

Reply: Thanks for the reviewer’s suggestion. We have revised the temperature gradient generation mechanism, and the function of the thermocapillary force in enhancing water evaporation and energy efficiency to make the concept of energy reutilizing property clearer. Corresponding revisions are added in the “Energy reutilizing through thermocapillary force” section of the **Discussion** part in the revised manuscript. Also, the description or definition of the energy reutilization property is added in the **Introduction** part of the revised manuscript in Line 20, Page 3: “Ascribing from the position-related structure inhomogeneity of the 3D structure, the generated water film on the evaporator surface displays thickness gradient. In addition, the input energy acquired by the bio-mimetic 3D evaporator system is related to the distance between the light source and the precise position, which results in the position-related energy utilization of the illumination. Cooperating with the position-related water evaporation on the bio-mimetic 3D evaporator induced by the structure inhomogeneity, the liquid film displays temperature gradient along the liquid film, indicating that the whole system unevenly utilizes energy. The simultaneously formed *Marangoni flow* lead to the water supplementation to the more vigorous evaporation site to enhance evaporation and energy efficiency, which finally leads to the energy reutilization property inside the water film”.

and in Line 18, Page 9: “The temperature gradient will further induce surface tension difference inside the liquid film, *i.e.*, the well-known *Marangoni* effect. It will provide a thermocapillary force (Equation 1) inside the liquid film to drive the liquid transportation from high temperature to low temperature (Equation 2), *i.e.*, from the bottom liquid film to the apex liquid film. As we have simulated and analyzed that the apex possesses higher steam flux both in darkness and under one sun illumination, water is thus continuously transported to the site with a higher evaporation rate, which can realize effective and continuous water evaporation. Therefore, the unevenly utilized input energy is further reutilized in the form of thermocapillary force, *i.e.*, the energy reutilization property of this system”.

9. **“Such high efficiency can be attributed to the reutilizing of solar energy through temperature gradient and the additional energy capture from the ambient environment.” Based upon my understandings, the sentence means that “Such high efficiency can be attributed to the reutilizing of solar energy through the formation of temperature gradient which allows for additional energy capture from the ambient environment.”**

Reply: Thanks for the reviewer’s suggestion. We have corrected corresponding sentence in Line 10, Page 14 in the revised manuscript as: “Comparing with other structures, the bio-mimetic 3D evaporator has the highest efficiency, which can be attributed to the reutilizing of solar energy through the formation of temperature gradient which allows for additional energy capture from the ambient environment”.

10. Additionally, the energy can be acquired from the surrounding environment increases owing to the decreased apex temperature environment increases?

Reply: Thanks for the reviewer's comment. The energy that can be acquired from the surrounding environment increases with the increased temperature difference between the apex temperature and the surrounding temperature. According to the reviewer's suggestion, we have revised the sentence "Additionally, the energy can be acquired from the surrounding environment increases owing to the decreased apex temperature" to "Additionally, the energy can be acquired from the surrounding environment increases owing to the increased temperature difference between the apex liquid film and the surrounding environment" in Line 6, Page 13.

11. When authors calculated the evaporation rate, what unit area did authors use? I mean projected area or real cone surface area? Probably, projected area, I guess. Why does the net evaporation rate (i.e, energy efficiency) decrease slightly when the H / D ratio increases from 1.4 to 2.0?

Reply: Thanks for the reviewer's question. The projected area is used for the calculation of energy efficiency. The value of efficiency is an average value of several calculated results, the slightly decrease of the value of energy efficiency is within the error range, which can be clarified from Figure 3i in the revised manuscript. The origin of error bars in measuring the water evaporation rate and calculating the energy efficiency has been added in the figure caption of Figure 3i and Figure 3j in the revised manuscript. Furthermore, we have checked the experimental data and found that with the increasing of H/D ratio from 1.4 to 2.0, both the average value of water evaporation rate in darkness and under one sun illumination increased, but the average value of net evaporation rate is slightly decreased, thus the average value of energy efficiency slightly decreased, but is within the error range.

Accordingly, we have added the "Projected area is used for the calculation of the mass change per area" in the **Characterization** part of the revised supplementary information.

12. Did authors consider the light modulation effect with respect to height especially during measuring the evaporation rate of high H/D ratio cone?

Reply: Thanks for the reviewer's question. In our experiment, we have used the top projected plane of 3D evaporators with different heights for measuring light intensity, as shown in **Figure R3**. The light intensity is equal to one sun at the top projected plane of the corresponding 3D structure. Therefore, with the same dimension of the projected area, the energy that 3D structures with different heights can acquire is normalized. We have added corresponding descriptions "For 3D evaporators with different H/D ratios, the top projected plane is utilized to normalize the light intensity" in the **Characterization** part of the revised supplementary information.

Figure R3. Scheme of the top projected area where the illuminated light intensity is equal to one sun.

13. The authors suggest a high energy efficiency of over 96%. To better highlight the reutilizing solar energy effect of the devices developed by the authors, it would be better to present and compare the energy efficiency of the 2D plane.

Reply: Thanks for the reviewer’s suggestion. With the average interfacial temperature of ~ 33.1 °C and the net evaporation rate of ~ 0.87 kg m⁻² h⁻¹ (the evaporation rate in darkness and under one sun illumination are 0.20 kg m⁻² h⁻¹ and 1.07 kg m⁻² h⁻¹, respectively), the calculated energy efficiency of the 2D plane is $\sim 59.2\%$. Corresponding descriptions “In the control experiment, the 2D plane has an average interfacial temperature of ~ 33.1 °C and the net evaporation rate of ~ 0.87 kg m⁻² h⁻¹, the energy efficiency of which is only $\sim 59.2\%$ ” has been added in the revised manuscript (Line 8, Page 14).

14. “Before use, the front side of 3D printed structures were plasma treated to endow the upper surface with hydrophilicity for water spreading.” For better understandings of hydrophilicity, please supply XPS result for upper surface in supporting information.

Reply: Thanks for the reviewer’s suggestion. We have conducted the X-ray photoelectron spectroscopy (XPS) characterizations of the upper surface of the bio-mimetic 3D evaporator before and after plasma-treatment (**Figure R4, Figure S18** in the revised supplementary information), and added corresponding results in the revised supplementary information. The peak value of the surface oxygen increases after plasma treatment, which proves that the surface content of -OH increases after plasma treatment. The addition of surface -OH will further lead to increased surface hydrophilicity.

Correspondingly, we have added, “After plasma treatment, surface hydrophilicity increases due to the increased amount of surface -OH. It can be clarified from the X-ray photoelectron spectroscopy (XPS) results as displayed in Figure S18, where the surface oxygen content increases” in the “**Post - 3D printing treatment**” section of the “**Methods**” part in the revised manuscript.

Figure R4 (Figure S18 in the revised supplementary information). XPS spectra of O 1s peaks of the upper surface of the bio-mimetic 3D evaporator before and after plasma treatment. After plasma treatment, the surface oxygen content increases, which proves the increase of the surface hydrophilicity originates from the surface -OH increase.

Responses to Reviewer # 2

In this study, the authors reported a novel 3D design for solar steam generation system which achieved high water evaporation rate and spatialized crystallization from high salinity water. At first glance, the solar water vaporizing performance is indeed excellent, especially for high salinity solutions. However, some of the explanations are not fully supported by solid evidences. There are still some other issues need to be studied in-depth. Considering the novelty and contributions to the field, I think the paper could be published in NC after these concerns are addressed. Below are the specific comments.

Reply: The authors thank the reviewer very much for the positive comments and the suggestions for improving the manuscript. According to the reviewer's comments, we have made revisions on our manuscript and addressed them one by one as below. The suggestions improve the quality of the revised manuscript. Thanks again for the reviewer's comments and suggestions.

- 1. The introduction needs to be improved. Particularly more details need to be added. The literature review failed in covering the state-of-art of studies in the solar steam field, which is critical for readers to understand the research gap this study aimed to address.**

Reply: Thanks for the reviewer's suggestion. We have revised the **Introduction** part by adding the references and corresponding discussions concerning the salt crystallization phenomenon during solar desalination process, including the strategies of inhibiting salt crystallization, reducing salt blockage or isolating salt crystals (**Page 2 to Page 3** in the revised manuscript). In addition, we have also summarized the representative references on solar steam generation and desalination with the variation of salinity, and compared the evaporation rate and energy efficiency among these references and our work. Corresponding data are added in **Table S1 and Figure S1** in the revised supplementary information.

- 2. In the first part of the discussion, the authors tried to illustrate the energy reutilisation through thermocapillary force. As described, the thermocapillary force can drive the liquid transportation from high temperature to low temperature. Nevertheless, how the thermocapillary force enhanced energy reutilisation and water evaporation? Moreover, the explanation (line 18-22, page 6) on the formation of temperature distribution under 1 sun or in the dark is not convincing. Detailed calculation or simulation may be helpful.**

Reply: Thanks for the reviewer's suggestion. According to the reviewer's comments, we have discussed the enhancement of water evaporation and energy efficiency by the thermocapillary force in detail, demonstrated the formation mechanism of temperature gradient in darkness and under one sun illumination, through adding new numerical simulations as the supporting evidences. Considering the 3rd question also concerns the function of thermocapillary force in enhancing water evaporation and energy efficiency, we answer this concern in the 3rd reply of the reviewer.

The generation of temperature gradient in darkness.

For dry bio-mimetic 3D structure (not putting into water) in darkness, the sidewall possesses a homogeneous temperature profile (Figure 3a in the revised manuscript) as there is no input or output of energy. After putting the evaporator into water, spontaneous water evaporation occurs after the generation of the liquid film on the structure, which results in the wet 3D structure. Ascribing from the position-related structure inhomogeneity on the bio-mimetic 3D evaporator, the generated liquid film displays a gradient thickness. The simulation result of the water/steam interface in darkness shows that the steam diffusion flux at the apex liquid film is larger than that at the bottom liquid film (**Figure R5a**, Figure 3h in the revised manuscript) based on the cooperation of the structure inhomogeneity and the liquid film thickness gradient, indicating a more vigorous water evaporation phenomenon at the apex. The simulation result is consistent with experimental result in Figure 3g (orange line). As is well-known, water evaporation absorbs energy and leads to the decrease of liquid surface temperature. The temperature of the apex liquid film, where more vigorous water spontaneous evaporation occurs, is thus lower than that at the bottom.

Figure R5. **a.** Numerical simulation result of the steam diffusion flux at the water/steam interface of the wet bio-mimetic 3D structure in darkness (Figure 3h in the revised manuscript). **b.** Numerical simulation result of the steam diffusion flux at the water/steam interface of the wet bio-mimetic 3D structure under one sun illumination (Figure 3f in the revised manuscript). The simulation process has been added in the **Methods** part of the revised manuscript.

The generation of the temperature gradient under one sun illumination.

We first discuss the temperature profile of the dry bio-mimetic 3D structure under one sun illumination. Light is perpendicularly illuminated to the projected plane of the bio-mimetic 3D structure. Due to the position-related structure inhomogeneity, the energy that the dry 3D structure can acquire from the illuminated light is different along the sidewall. With a closer position from the light source, higher energy intensity can be acquired at the apex. Cooperating with a smaller surface area at the apex, it will lead to a higher apex structure temperature. Thus, the energy that the bio-mimetic 3D structure can acquire is position-related, *i.e.*, the position-related utilization feature of the input energy. The analyses agree well with the experimental measurement by the infrared camera, as shown in **Figure 3b** in the revised manuscript, which further proves our explanation on the generation of temperature gradient of the dry 3D structure under one sun illumination.

We then put the bio-mimetic 3D structure onto a water bath. Water film spontaneously formed on the bio-mimetic 3D structure with a thickness gradient from the apex to the bottom, which is not influenced by the illumination. Combining with the liquid film thickness data and the temperature gradient data of the dry 3D structure as stated above, water evaporation phenomenon at the apex liquid film is thus more vigorous than that at the bottom film, which leads to the enhanced position-related water evaporation under one sun illumination comparing with the situation in darkness. The simulation result agrees well with the above statement that under one sun illumination, the steam flux at the apex liquid film is higher than the bottom (**Figure R5b**, Figure 3f in the revised manuscript). As water evaporation will absorb energy and take heat away, it will in turn decrease the surface temperature of the liquid film. Therefore, the temperature distribution along the liquid film is determined by the competition between the position-related energy absorption and transfer inside the system, and the position-related evaporation of the liquid film.

The temperature profile image is the competition result of the processes mentioned above, which is also position-related obviously. Comparing with the source liquid temperature, the apex liquid film temperature is decreased, while the bottom liquid film temperature is increased. It indicates that for the apex liquid film the evaporation dominates, while at the bottom liquid film the energy absorption and transfer from the bottom structure dominates. In conclusion, the formulation of temperature gradient in darkness and under one sun illumination can be attributed to the position-related competition among energy absorption and transfer inside the system and the position-related evaporation along the water film.

The function of thermocapillary force in enhancing water evaporation and energy efficiency.

The function of thermocapillary force in enhancing water evaporation and energy efficiency has been explained in detail in the 3rd question of the reviewer with new control structure and new experimental results.

Correspondingly, the “**Energy reutilizing through thermocapillary force**” section in the **Discussion** part is reorganized based on the above analyses in the revised manuscript.

3. In this paper, the enhanced water evaporation performance mainly due to the efficient dark evaporation, which was reported $0.84 \sim 1.17 \text{ kg m}^{-2} \text{ h}^{-1}$ for the 3D evaporator. This enhancement can be well explained by the energy harvesting from the environment (Joule, 2018, 2, 1331-1338). So how to prove and demonstrate the real benefits of thermocapillary force? More experimental results and explanations are required.

Reply: Thanks for the reviewer’s comment. In order to demonstrate the real benefits of thermocapillary force in enhancing water evaporation and energy efficiency, we designed and prepared a 3D columnar structure for comparison. As displayed in **Figure R6a (Figure S12** in the revised supplementary information), the sidewall and upper surface of the 3D columnar structure is composed of 25 grooves of micro-cavity arrays without asymmetry. It possesses the same H/D ratio (0.7) and projected area with the bio-mimetic 3D structure used in the manuscript. Due to the columnar structure without asymmetry, micro-cavities along each groove of the sidewall possesses the same dimension

without gradient. The top surface of the 3D columnar structure is designed with 25 radially patterned asymmetric grooves composed of micro-cavity arrays with the same height of the micro-cavity on the sidewall for sustaining the continuous liquid film. Otherwise, water will dewet on the top surface during water evaporation, which will decrease the efficient water/structure contact area. The liquid film generated on the 3D columnar structure is homogeneous both on the sidewall and the top surface without thickness gradient. The 3D columnar structure is also prepared from the size-dependent resin refilling induced 3D printing through employing the composite resin in Figure S3 with the same post 3D printing treatment procedure.

Figure R6. **a.** Scheme of the detailed morphology of the 3D columnar structure (Figure S12 in the revised supplementary information). The 3D columnar structure possesses 25 grooves of micro-cavity arrays along the sidewall of the columnar structure. The two structures possess the same projected area used for calculating the water evaporation rate and the same H/D ratio. **b.** Temperature profiles along the bio-mimetic 3D structure and 3D columnar structure surface in darkness. Red and black lines represent temperature profiles along the sidewall of the bio-mimetic 3D structure and the 3D columnar structure, respectively. **c.** Temperature profiles along the bio-mimetic 3D structure and the 3D columnar structure surface under one sun illumination. Red and black lines represent temperature profiles along the sidewall of the bio-mimetic 3D structure and the 3D columnar structure, respectively. Insets are corresponding infrared images showing the temperature distribution at the equilibrium state on the bio-mimetic 3D structure and the 3D columnar structure under one sun illumination.

In darkness, the water evaporation rate on the 3D columnar structure is lower than that on the bio-mimetic 3D structure. The temperature difference between the top and bottom liquid film on the 3D columnar structure is smaller than that on the bio-mimetic 3D structure, as shown in Figure R6b (red and orange lines in Figure 3g in the revised manuscript). Consequently, smaller thermocapillary force is generated on the 3D columnar structure than the bio-mimetic 3D structure. However, the average surface temperature of the 3D columnar structure is lower than that on the bio-mimetic 3D structure, which means that the energy can be acquired from the surrounding environment is also less. Therefore, the contribution of thermocapillary force in enhancing the water evaporation in darkness is hard to distinguish for the two structures.

Under one sun illumination, the average surface temperatures of the two structures are almost the same, which means that the energy acquired from the surrounding environment can be considered as the same (Table R1, Table S2 in the revised supplementary information). However, the trend of temperature profiles on the two structures is different (Figure R6c, red and orange lines in Figure 3e

in the revised manuscript). On the 3D columnar structure, the top surface temperature is higher than the bottom, whose trend is contrary to the bio-mimetic 3D structure. It will lead to the opposite direction of the thermocapillary force on the 3D columnar structure compared with the bio-mimetic 3D structure. The direction of water flow induced by thermocapillary force is thus from the top surface along the sidewall to the bottom on the 3D columnar structure, which is also contrary to the supplementation direction of liquid from the source water. Consequently, the water supplementation from the source to the evaporator surface is hindered during the continuous evaporation process, which finally leads to the reduced water evaporation rate comparing with the bio-mimetic 3D evaporator.

Accordingly, we have incorporated the temperature profile data of the 3D columnar structure in **Figure 3e and Figure 3g** in the revised manuscript, added the solar steam generation performance of the 3D columnar structure as **Table S2** in the revised supplementary information, and added corresponding analyses in the revised manuscript.

Table R1 (Table S2 in the revised supplementary information). Comparison of the solar-driven water evaporation parameters of the bio-mimetic 3D evaporator and the 3D columnar evaporator.

	Bio-mimetic 3D Evaporator	3D Columnar Evaporator
Top Temperature (°C)	33.6	35.3
Bottom Temperature (°C)	21.0	20.2
Average Temperature (°C)	28.9	30.5
Water Evaporation in Darkness (kg m ⁻² h ⁻¹)	0.84	0.62
Water Evaporation under One Sun Illumination (kg m ⁻² h ⁻¹)	2.28	1.78
Energy Efficiency (%)	97.5	72.0

Notes: The temperatures and the energy efficiency are the measured or calculated data under one sun illumination.

4. The spatialized salt crystallization is not well explained. The reason why crystallization happened at the apex (Line 16-18 on page 10) might not because of its faster evaporation. In addition, the statement “the inhomogeneous temperature distribution also leads to spatialized salt crystallization feature on the 3D evaporator” (line 12, page 3) is not well proved in the results and discussions. Actually, similar results have been reported (Environmental Science & Technology, 2018, 11822-11830; Energy & Environmental Science, 2019, 12, 1840-1847). Suggest the authors read these papers and improve their discussions on spatialized salt crystallization.

Reply: Thanks for the reviewer’s comment. According to the reviewer’s suggestion, we have revised the “**Localized crystallization and salt accumulation for high-efficiency and sustainable desalination**” section of the **Discussion** part referring the above mentioned two papers, and we have also cited these two papers in the revised manuscript. The sentence “the inhomogeneous temperature distribution also leads to spatialized salt crystallization feature on the 3D evaporator” has been corrected to “the liquid film thickness gradient and the position-related water evaporation along the liquid film also leads to the salt concentration gradient and localized salt crystallization feature on the 3D evaporator” in the revised manuscript (Line 9, Page 4).

As analyzed in Figure 3 in the revised manuscript, position-related water evaporation occurs along the liquid film surface, where the apex liquid film evaporates more vigorously than the bottom liquid film both in darkness and under one sun illumination. Correspondingly, the position-related water evaporation and the thickness gradient along the liquid film function cooperatively to the formation of salt concentration gradient along the liquid film, *i.e.*, position-related salt concentration along the liquid film, where the salt concentration at the apex liquid film is higher than that at the bottom liquid film. The apex liquid film is thus easier to reach the critical crystallization concentration comparing with the bottom. Therefore, the upper the position on the bio-mimetic 3D structure, the higher the salt concentration and the higher tendency for salt crystallization. In addition, due to the continuous liquid film formed between the crystal and the bio-mimetic 3D structure, the crystal which is not formed at the apex can also move along with the upward supplementing water flow from the bottom to the apex. Therefore, in addition to salt crystallization preference at the apex structure, salt is also preferred to move to the apex position, which finally leads to the accumulation of the crystallized salt at the apex. Accordingly, we have also revised the subtitle of “Spatialized crystallization for high-efficiency and sustainable desalination” to “Localized crystallization and salt accumulation for high-efficiency and sustainable desalination” in the revised manuscript.

Responses to Reviewer # 3

In this paper, the authors report 3D printing interfacial solar evaporation structure. The structure achieved high evaporation efficiency. The authors invoked Marangoni effect to explain the high efficiency. Overall, the results are interesting. However, I do not feel the paper contains enough new science or technology advances that warrant its acceptance in Nature Communication. My reasons and comments are detailed below.

Reply: The authors thank the reviewer very much for the interests on our result. The comments from Reviewer #3 mainly focuses on the energy transfer mechanism and the experimental details of this work, failing to clearly point out the science or technology absence based on comments listed below. According to the reviewer's comments, we have made revisions on our manuscript and addressed them one by one as below, and hope the revisions can clearly demonstrate the mechanism and advances of our work. The suggestions help to improve the clear description of the mechanism part and to improve the quality of the revised manuscript. Thanks for the reviewer's suggestions.

1. There are many papers published on this topic, and quite high efficiency has been reported. Some at comparable level.

Reply: Thanks for the reviewer's comment. We want to emphasize that our work not only reaches high efficiency at high salinity, but also introduces a localized salt accumulation feature with salt removing property for sustainable applications.

As the reviewer mentioned, there are many papers published on this topic, because solar desalination is considered to be an efficient way to acquire clean water with solar energy as the only energy input. However, in these published papers, high energy efficiency has only been achieved during solar evaporating of pure water or low salinity water (with salt concentration ≤ 10 wt%, the yellow shadow region in **Figure R7**, **Figure S1** in the revised supplementary information). How to increase the evaporation rate and the energy efficiency under high salinity is of great importance for practical usage (>10 wt%, the grey shadow in **Figure R7**, **Figure S1** in the revised supplementary information). In practical application, with the progressing of the desalination process, the salt concentration around the evaporator increases. The increase of salt concentration will further lead to salt deposition on the evaporator, which is an obstacle for long-term usability for current solar desalination filed. Though salt blockage can be inhibited through manually introduced salt-rejection pathways, the efficiency is reduced by the high salt content (**Table R2**, the salt rejecting properties). In addition, the increased salt concentration during solar-driven evaporation process inevitably results in salt crystallization on the evaporator surface and further inhibits the solar desalination process. Moreover, the complete removal of the deposited salt from the evaporator for long-term utilization has been not solved (**Table R2**, the salt removing properties).

To clearly demonstrate the advances of our work, we have summarized the representative references on solar steam generation and desalination with the variation of salt concentration, and compared the evaporation rate and energy efficiency under one sun illumination among these references. The

comparison is summarized into figures and tables, which have been added as Figure S1 and Table S1 in the revised supplementary information. Comparing with previous works, our developed bio-mimetic 3D evaporators achieved records of high evaporation rate and high energy efficiency even under high salinity, with crystalized salt free-stands at the apex without contaminating the 3D evaporator for long-term utilization. In addition, the **Introduction** part has been revised through adding new references and corresponding descriptions to clearly demonstrate the current state of the solar desalination field and the advances of our work.

Figure R7 (Figure S1 in the revised supplementary information). Comparison of solar desalination performance of representative references with our work, including the evaporation rate and energy efficiency under one sun illumination with the variation of salt concentration.

Table R2 (Table S1 in the revised supplementary information). Summary of representative references on solar-driven desalination with the variation of salt concentration¹⁻¹⁶.

Ref.	Salt Concentration (wt%)	Mass Change (kg m ⁻² h ⁻¹)	Energy efficiency (%)	Salt removing capability	Salt rejecting property	Notes
1	0	3.2	~94	/	/	Reduced water vaporization enthalpy, with evaporation rate of 3.2 kg m ⁻² h ⁻¹ , no data on high-salinity samples
2	0	1.62	100	/	/	/
3	~ 3.5	1.25	91	/	/	Commercial synthetic seawater
4	~ 4.5	1.97	91.7	/	/	seawater sample from Nanhai Sea
5	10	1.28	78.5	/	√	Brine sample: NaCl solution
6	15	0.8	57	/	√	Brine sample: NaCl solution
7	15	0.5	~80	/	/	Brine sample: NaCl solution
8	20	1.04	75.0	/	√	Brine sample: NaCl solution
9	25	1.36	88.4	/	/	Brine sample: NaCl solution
10	26	1.42	81.2	√	/	Brine sample: NaCl solution; Localized crystallization, salt falls down due to gravity
11	0	1.3	72	/	/	20wt% NaCl solution capable, without data
12	0	2.12	91.5	/	/	The data are measured under reduced pressure (0.25 atm)
13	1.19	1.99	/	/	/	Energy efficiency for pure water is ~100%, 1.19 wt% is the solid content inside the wastewater

14	3.5	2.5 kg m ⁻² day ⁻¹	56±2.5	/	√	Brine sample: NaCl solution, with NaCl solid on the evaporator
15	~ 10	2.5	~95	/	√	Reduced water vaporization enthalpy, Dead Sea water sample
16	21	1.94	89.9	/	/	Brine sample: NaCl solution
Our work	25	2.24 (H/D 0.7) 2.6 (H/D 1.4)	>96	√	/	Brine sample: NaCl solution; Localized crystallization, free standing salt

2. Although the reported structure is clearly 3D, it is difficult to say other porous structures are not 3D. In fact, these structures achieved high efficiency because the large surface area.

Reply: Thanks for the reviewer's comment. The mechanism and function of the bio-mimetic 3D structure are different from other porous structures.

First, the employment of other porous structure aimed to increase the effective water/structure contact interface to enhance water evaporation. The employment of 3D structure in our work aimed not only to increase the surface roughness, but also to introduce position-related structure inhomogeneity and corresponding liquid film thickness gradient to enhance water evaporation.

Second, our work possesses a different enhancing mechanism. For other porous structures, water spreads uniformly and forms a homogeneous liquid/structure interface. Our work proposes an inhomogeneous liquid/structure interface, which further leads to the energy reutilization through thermocapillary force and the enhanced water evaporation and energy efficiency.

Third, the surface-distributed micropores in our system can sustain a continuous liquid film between the crystalized salt and the bio-mimetic 3D structure, which leads to a free-standing and porous salt for easy removing and long-term utilization comparing with other porous structures.

3. Comparable efficiency could be reached using much simpler and cheaper structures. It is unlikely such 3D fabricated structures are cost competitive for applications.

Reply: Thanks for the reviewer's comment. The contribution of our work is that we have demonstrated an asymmetric 3D structure that can serve as an efficient solar steam generation and desalination device for long-term utilization. With this 3D evaporator, the overall performance concerning water evaporation rate, energy efficiency, and desalination property is superior to other systems as displayed in **Table R2** (**Table S1** in the revised supplementary information). We have employed 3D printing in our work to prepare the 3D evaporator, but the fabrication strategy to this kind of inhomogeneous structure is not limited to 3D printing. Whereas, the practical application for solar desalination is another issue, which should not only consider the cost for preparing the evaporator, but also the life cycle, the capability to maintain the stable property during long-term utilization and the cost for maintenance, *etc.* In addition, with the rapid development in 3D printing technology, the fabrication efficiency has been significantly increased and the fabrication cost has been significantly decreased. It is easy for researchers or 3D printing enthusiasts to build a 3D printing prototype now. Therefore, 3D printing is becoming more and more competitive as a manufacturing strategy.

The above are nontechnical but support my lack of enthusiasm for the paper. Technical comments are listed below.

- 1. The claimed high efficiency does not seem to commensurate with the optical measurements. It seems that the absorptance is around 90%. The highest report efficiency is over 98%. If one calculates the efficiency based on absorbed heat, it would be higher than 100%. This is not explained.**

Reply: Thanks for the reviewer's comment. As indicated in the manuscript, the average surface temperature is lower than the surrounding environment (Figure 3e - Figure 3j in the revised manuscript), energy can also be acquired from the surrounding environment based on the references (Joule, 2018, 2, 1331-1338; Joule, 2018, 2, 1171-1186), which has been stated in the manuscript (Line 2, Page 11) as: "Based on the previous investigations^{11,12}, energy can also be directly collected from the surrounding environment, which contributed cooperatively with thermocapillary force to enhance solar-driven water evaporation". Therefore, the input energy source not only includes the illuminated light source, but also includes the energy acquired from the surrounding environment, which contributed cooperatively to the high energy efficiency.

- 2. Heat transfer mechanisms during the continuous evaporation process and how the illumination of the sunlight boosts the evaporation are not clear to me. At the wavelengths around 10 micron, the penetration length of water is around 20 micron. From Fig.4e, it seems that the water film thickness near the apex is on the order of 100 micron. Thus, the temperature distribution measured by the infrared camera is the average temperature distribution near the liquid-vapor interface. The question I have is what the temperature distribution of the wall temperature of the conical structure is under one sun illumination. My understanding is that the wall temperature at the apex is higher than that at the bottom. Since the water is transparent to the sunlight, the conical structure is a wet fin with varying cross-section under the uniform heat flux. The water film thickness is thinner near the apex, which is equivalent to the higher heat transfer coefficient near the apex, which leads to the lower liquid-vapor interface temperature near the apex than that at the bottom under one sun illumination. However, it seems that the higher wall temperature at the apex due to the sunlight and the larger heat transfer coefficient near the apex due to a thinner liquid film are competing each other in creating the inverted temperature distribution of the liquid-vapor interface. In other words, if the increasing heat transfer coefficient towards the apex is overridden by the increasing wall temperature towards the apex, the temperature distribution of the liquid-vapor interface may be inverted (higher temperature at the apex). I would like the authors to more clearly explain how the incident sunlight on the surface of the conical structure boosts the evaporation rate.**

Reply: Thanks for the reviewer's comment. Macroscopically, as water can spontaneously evaporates, the energy of the incident sunlight can be absorbed by the bio-mimetic 3D structure as thermal energy, which can be further transferred to the water film to enhance or boost the spontaneous water evaporation.

For the dimension of the liquid film, the thinnest part of the liquid film at the apex is $\sim 15 \mu\text{m}$ and the thickest part at the bottom is $\sim 1500 \mu\text{m}$ through analyzing the reconstructed Micro-CT image in Figure 2g - Figure 2h. Accordingly, we have added the description of the liquid film thickness in the revised manuscript, Line 10, Page 6 of the revised manuscript: “a 3D water film with thickness inhomogeneity along the sidewall is spontaneously formed, where the apex liquid film ($\sim 15 \mu\text{m}$ the thinnest) is thinner than the bottom liquid film ($\sim 1500 \mu\text{m}$ the thickest), as displayed in Figure 2g - Figure 2h”.

In addition, the energy transfer from the 3D structure to the liquid film is effective, which can be reflected by the surface temperature of the thickest bottom liquid film. Even with the film thickness of $\sim 1500 \mu\text{m}$ at the bottom, the surface temperature of the liquid film can also be heated to $\sim 34.5 \text{ }^\circ\text{C}$ (red line, Figure 3e in the revised manuscript), indicating the effective energy transfer from the structure to the liquid film. As displayed in **Figure R8** (Figure S10 in the revised supplementary information), each groove of water is surrounded by the three connected sidewalls of corresponding groove, where annular heating from the three directions of the groove structure occurs (red arrows in the inset of Figure R8, Figure S10 in the revised supplementary information). Therefore, the designed groove structure, and the contacting mode of water with corresponding grooves contributed to the effective energy transfer from the 3D structure to the liquid film. Corresponding descriptions are added in the revised manuscript in Line 19, Page 8 as: “It is worth mentioning that the thickest bottom liquid film can be heated to $\sim 34.5 \text{ }^\circ\text{C}$ after reaching the equilibrium state, indicating the sufficient energy transfer from the bio-mimetic 3D structure to the liquid film, basing on the designed groove structure and the contacting mode of water with corresponding grooves, as displayed in Figure S10”.

Figure R8 (Figure S10 in the revised supplementary information). Scheme of the contact mode of each groove of water with corresponding groove on the bio-mimetic 3D evaporator. Red arrows indicate the energy transfer directions from the 3D structure to corresponding water groove.

For the energy transfer inside the system and the generation mechanism of temperature gradient, to some extent, we agree with the reviewer’s understanding. However, the reviewer concerns the energy transfer from the input light to the 3D structure, the energy transfer from the structure to the liquid film, but ignores the energy transfer by water evaporation during the solar steam generation process. Water evaporation will absorb energy and take heat away, which will in turn decrease the liquid film temperature. Supposing that water is continuously heated by the structure without evaporation, the thinner apex liquid film, which is heated by the apex structure with higher temperature, is easier to

reach a higher temperature than the bottom liquid film. The situation is consistent with the reviewer's description. Whereas, for a thinner liquid film, it not only possesses a higher heat transfer coefficient, but also a higher tendency to evaporate. Therefore, the surface temperature of the liquid film is the result of the competition between the heat transfer from the structure to increase the liquid temperature, and the evaporation of water which will decrease the liquid temperature.

The mechanism of the incident sunlight on the surface of the bio-mimetic 3D evaporator to further enhance the water evaporation is based on the cooperation of the thermocapillary force and the energy capture from the surrounding environment, which has been revised in the **Discussion** part of the manuscript, with simulation results as supporting evidences for more clear description and easier understanding.

If the conical part is shielded and if the sunlight is incident only on the water bath around the conical structure, does that boost the evaporation further since you create a greater temperature gradient between the bottom and the apex?

Reply: Thanks for the reviewer's comment. The situation mentioned by the reviewer is a more complex system than just stated. Because the light is not directly illuminated on the tested structure or interface, but the surrounding water bath. The mass loss of such a system is at least composed by three parts: 1) the mass loss of the water bath due to illumination; 2) the mass loss of the 3D evaporator in darkness; 3) the additional mass loss of the 3D evaporator due to increased water source temperature. It is hard to distinguish the contribution of each component for the entire mass loss. Also, the sufficient surface area used for calculating the water evaporation rate is hard to define for such a complicate system.

Even though, we have tried to measure the temperature profile along the 3D evaporator and the water bath with light illuminating from one side of the water bath rather than surrounding the 3D structure, as shown in **Figure R9a**. Because the infrared camera can only measure the surface temperature, illuminating light around the structure will influence the measured temperature on the 3D evaporator surface by infrared camera. The illuminated light from one side will lead to the heating of the water bath. At the side of water bath under illumination, the water surface temperature is higher than the side without illumination. In addition, the temperature of liquid film on the 3D structure is synchronously heated, which can be due to the sufficient water supplementation from the water bath to the 3D evaporator.

As shown in **Figure R9b**, the temperature difference between the apex liquid film and the bottom liquid film at the side without illumination (left side, red line in **Figure R9b**) remains the same with the condition of 3D evaporator in darkness (black line, **Figure R9b**), while the temperature difference between the apex liquid film and the bottom liquid film at the side with illumination becomes larger (right side, red line in **Figure R9b**). Therefore, the thermocapillary force at the side with illumination will be increased based on the mechanism proposed by our work, which will lead to enhanced water evaporation. Whereas, the concrete contribution of the thermocapillary force in enhancing water evaporation is hard to calculate for such a complex system.

Figure R9. Comparison of temperature profiles among 3D evaporator in darkness, 3D evaporator under one sun illumination and illuminating one side of the water bath beside the 3D evaporator. **a.** Scheme of the experimental detail of illuminating one side of the water bath beside the 3D evaporator. **b.** Temperature profiles along the 3D structure under one sun illumination (blue line), along the 3D evaporator in darkness (black line) and along the 3D evaporator with illumination of the water bath beside the 3D evaporator.

3. The authors should explain the side of the container used to test the device. The tested sample is small, and there could be lots of parasitic evaporation from sides. They need to present details of experiments and how various parasitics are considered.

Reply: Thanks for the reviewer's comment. According to the reviewer's suggestion, we have added the detailed experimental apparatus used for the open system and the closed system in the revised supplementary information.

First, for the open system, the employed container is a U-shaped glass tube, as shown in **Figure R10a** (Figure S7 in the revised supplementary information). The left side of the U-shaped glass tube is straight (left side, Figure R10a), while the opening of the right side is higher than the left side. Besides, the right side possesses a spherical protrusion at the same height with the opening of the left side (right side, Figure R10a). The diameter of the spherical protrusion is about 2.5 times the diameter of the straight left side. For an open system, the heights of the liquid surfaces on both sides should be the same ascribing from the property of the U-shaped tube. Therefore, extracting liquid from one side, liquid will be supplemented from the other side until reaching the same liquid height on both sides (Figure R10b - Figure 10d). Calculating from the solar steam generation rate of the bio-mimetic 3D evaporator, the liquid height can be kept almost unchanged for at least 4 hours ascribing from the large spherical protrusion on the right side. The self-float 3D evaporator will not fall inside the tube during the measuring process. Therefore, the effective solar steam generation surface of the bio-mimetic 3D evaporator in the open system is considered to be in free contact with the surrounding air and is not sheltered or influenced by the sidewall of the container. In addition, the opening of the right side of the U-shaped tube is filled with hydrophobic-treated cotton to inhibit water evaporation from the right side.

Figure R10 (Figure S7 in the revised supplementary information). a. Optical image of the experimental apparatus for measuring the evaporation rate in open system. b - d. Optical captures of extracting liquid from the left side of the U-shaped tube. The liquid height of the left side remains almost unchanged after extracting liquid from the left side.

Second, for the closed system, the container is a polyethylene coverless box with the dimension of 11.8 cm × 11.8 cm, while the dimension of the printed 3D evaporator array is 10.4 cm × 10.4 cm, whose intervals are designed to fill with flat surfaces. The detailed dimensional parameters are displayed in **Figure R11 (Figure S17 in the revised supplementary information)**. The supplementation of brine water is through the inlet, and the water collecting rate is calculated through the amount of water collected from the outlet per three hours. The closed system is a much more complex system compared with the open system, many parasitics, *e.g.*, the increased humidity inside the condenser, the condensation of vapor on the top and sidewall of the condenser, the fall down of the condensed water from the top and sidewall of the condenser, and the supplementation velocity of source water from the inlet, *etc.*, may influence the water evaporation rate. Therefore, for the closed system, we measured the water collecting rate rather than the water evaporation rate.

Figure R11 (Figure S17 in the revised supplementary information). Optical images and corresponding geometric parameters of the closed system used for measuring the water collection rate.

Accordingly, we have added corresponding figures and descriptions of the experimental apparatus used for the open system (Figure S7) and the closed system (Figure S17) in the revised supplementary information.

References

1. Zhao, F. *et al.* Highly efficient solar vapour generation via hierarchically nanostructured gels. *Nat. Nanotechnol.* **13**, 489-495 (2018).
2. Li, X. *et al.* Enhancement of Interfacial Solar Vapor Generation by Environmental Energy. *Joule* **2**, 1331-1338 (2018).
3. Shi, L. *et al.* Multi-functional 3D honeycomb ceramic plate for clean water production by heterogeneous photo-Fenton reaction and solar-driven water evaporation. *Nano Energy* **60**, 222-230 (2019).
4. Cui, L. F. *et al.* High Rate Production of Clean Water Based on the Combined Photo-Electro-Thermal Effect of Graphene Architecture. *Adv. Mater.* **30**, 1706805 (2018).
5. Xu, N. *et al.* A water lily–inspired hierarchical design for stable and efficient solar evaporation of high-salinity brine. *Sci. Adv.* **5**, eaaw7013 (2019).
6. He, S. M. *et al.* Nature-inspired salt resistant bimodal porous solar evaporator for efficient and stable water desalination. *Energy Environ. Sci.* **12**, 1558-1567 (2019).
7. Finnerty, C., Zhang, L., Sedlak, D. L., Nelson, K. L. & Mi, B. X. Synthetic Graphene Oxide Leaf for Solar Desalination with Zero Liquid Discharge. *Environ. Sci. Technol.* **51**, 11701-11709 (2017).
8. Kuang, Y. *et al.* A High - Performance Self - Regenerating Solar Evaporator for Continuous Water Desalination. *Adv. Mater.* **31**, 1900498 (2019).
9. Shi, Y. *et al.* Solar Evaporator with Controlled Salt Precipitation for Zero Liquid Discharge Desalination. *Environ. Sci. Technol.* **52**, 11822-11830 (2018).
10. Xia, Y. *et al.* Spatially isolating salt crystallisation from water evaporation for continuous solar steam generation and salt harvesting. *Energy Environ. Sci.* **12**, 1840-1847 (2019).
11. Xu, W. C. *et al.* Flexible and Salt Resistant Janus Absorbers by Electrospinning for Stable and Efficient Solar Desalination. *Adv. Energy Mater.* **8**, 1702884 (2018).
12. Li, W., Li, Z., Bertelsmann, K. & Fan, D. E. Portable Low-Pressure Solar Steaming-Collection Unisystem with Polypyrrole Origamis. *Adv. Mater.*, 1900720 (2019).
13. Shi, Y. S. *et al.* A 3D Photothermal Structure toward Improved Energy Efficiency in Solar Steam Generation. *Joule* **2**, 1171-1186 (2018).
14. Ni, G. *et al.* A salt-rejecting floating solar still for low-cost desalination. *Energy Environ. Sci.* **11**, 1510-1519 (2018).
15. Zhou, X. Y., Zhao, F., Guo, Y. H., Zhang, Y. & Yu, G. H. A hydrogel-based antifouling solar evaporator for highly efficient water desalination. *Energy Environ. Sci.* **11**, 1985-1992 (2018).
16. Liu, Z. *et al.* Continuously Producing Watersteam and Concentrated Brine from Seawater by Hanging Photothermal Fabrics under Sunlight. *Adv. Funct. Mater.*, 1905485 (2019).

Reviewers' comments:

Reviewer #1 (Remarks to the Author):

Those are my additional suggestions for further improvement after reading authors' corrections and answers;

4. Authors mentioned "It should be noted that the water upward moving velocity on the 3D structure is much larger than that on porous filter paper (Figure S5)." This means probably faster not larger. By the way, Figure S5 c displays wrong indication for filter paper (red) and mimic (black).

Reply: Thanks for the reviewer's comment. The sentence "It should be noted that the water upward moving velocity on the 3D structure is much larger than that on porous filter paper (Figure S5)" has been revised to "It should be noted that the water upward moving velocity on the bio-mimetic 3D structure is much faster than that on porous filter paper (Figure S6)" (Line 3, Page 6 in the revised manuscript). Additionally, we have revised the indications for filter paper and the bio-mimetic

Additional suggestion: Some brief discussion may be added to point out the abruptness of the curve for bio-mimetic structure. In the discussion, authors could answer the question: "Why does the shape of the curve for Bio-mimetic structure looks step-like and that for filter-paper is smooth?" I think it is important that the point of authors' paper is in the uniqueness of the structure. Thus, stressing different and beneficial water transport would strengthen the paper.

I think the answer would be in the difference of the water transport mechanism.

7. higher structure temperature > need other expression

Reply: Thanks for the reviewer's suggestion. The description of "higher structure temperature" concerns the generation of the temperature gradient of the dry 3D structure and water filled wet 3D structure under one sun illumination. In order to make the temperature gradient generation mechanism clearer, we have revised the "Energy reutilizing through thermocapillary force" section in the Discussion part with new descriptions and the addition of new simulation results.

Additional suggestion: The expression "higher structure temperature" is somehow confusing and may be understood as the temperature of the higher structure. Please rephrase the expression to avoid confusion. Higher structure temperature -> Higher temperature of the structure.

14. "Before use, the front side of 3D printed structures were plasma treated to endow the upper surface with hydrophilicity for water spreading." For better understandings of hydrophilicity, please supply XPS result for upper surface in supporting information.

Reply: Thanks for the reviewer's suggestion. We have conducted the X-ray photoelectron spectroscopy (XPS) characterizations of the upper surface of the bio-mimetic 3D evaporator before and after plasma-treatment (Figure R4, Figure S18 in the revised supplementary information), and added corresponding results in the revised supplementary information. The peak value of the surface oxygen increases after plasma treatment, which proves that the surface content of -OH increases after plasma treatment. The addition of surface -OH will further lead to increased surface hydrophilicity.

Correspondingly, we have added, "After plasma treatment, surface hydrophilicity increases due to the increased amount of surface -OH. It can be clarified from the X-ray photoelectron spectroscopy (XPS) results as displayed in Figure S18, where the surface oxygen content increases" in the "Post - 3D printing treatment" section of the "Methods" part in the revised manuscript.

Additional question: What is the meaning of the "peak value of the surface oxygen"? Do the authors imply the height or area of O1s peak? And if so, how can those increases be evidence of the increases of the -OH groups? It is advisable to provide proper peak deconvolution results to clearly show changes in the surface chemistry. Furthermore, it will give more clarity if C1s peak is provided and sufficiently analyzed as well.

Reviewer #2 (Remarks to the Author):

All comments from reviewers have been addressed carefully. The authors have also conducted additional experiments and simulation to support their conclusions. I would like to recommend a publication in NC although a few minor issues are still needed to be addressed:

1. More details about the stimulation are needed to be provided in the supporting materials, such as the mesh, boundary conditions, physical parameter and validation.

2. The evaporation rates reported in Table R1 and Table S2 are different. Please double check.

3. In the section of “The generation of the temperature gradient under one sun illumination”, the authors state, “Due to the position-related structure inhomogeneity, the energy that the dry 3D structure can acquire from the illuminated light is different along the sidewall. With a closer position from the light source, higher energy intensity can be acquired at the apex. Cooperating with a smaller surface area at the apex, it will lead to a higher apex structure temperature.”

It is true for a solar simulator. However, in real situations, this distance-related energy intensity may be negligible. In addition, what do you mean by “the apex has smaller surface” and why it will lead to a higher apex structure temperature? Please give more description.

Responses to Reviewer # 1

Those are my additional suggestions for further improvement after reading authors' corrections and answers;

Reply: We greatly appreciate the reviewer for the valuable additional suggestions. According to the reviewer's comments, we have carefully revised the manuscript. We hope that the revised manuscript would be suitable for publication in *Nature Communications*.

- 1. 4. Authors mentioned "It should be noted that the water upward moving velocity on the 3D structure is much larger than that on porous filter paper (Figure S5)." This means probably faster not larger. By the way, Figure S5 c displays wrong indication for filter paper (red) and mimic (black).**

Additional suggestion: Some brief discussion may be added to point out the abruptness of the curve for bio-mimetic structure. In the discussion, authors could answer the question: "Why does the shape of the curve for Bio-mimetic structure looks step-like and that for filter-paper is smooth?" I think it is important that the point of authors' paper is in the uniqueness of the structure. Thus, stressing different and beneficial water transport would strengthen the paper.

I think the answer would be in the difference of the water transport mechanism.

Reply: We thank the reviewer very much for the additional comments and suggestion. The water transport inside the filter paper is based on the porous structures induced capillary wicking, whose speed is inhibited by the small porous structures of the filter paper. However, water transportation on our bio-mimetic structure is a kind of surface water transport that based on the continuous filling of the micro-cavities array along the sidewall. The micro-cavity, which is the standard structure of the peristome surface of the pitcher plant, allows continuous and inward liquid transport. Liquid spreads along the micro-cavity with an accelerating speed as the precursor approaching the apex of the cavity and overflow into the next micro-cavity to repeat the spreading process, leading to a step-like transport behavior. Comparing with the water inside filter paper, water shows an ultra-fast water transport speed on the biomimetic structure.

According to the reviewer's suggestion, we have revised the sentence from "It should be noted that the water upward moving velocity on the bio-mimetic 3D structure is much faster than that on porous filter paper (Figure S6)" to "It should be noted that the water upward moving velocity on the bio-mimetic 3D structure based on the micro-cavity induced water continuous and inward liquid transportation, is much faster than that on porous filter paper originated from porosity induced capillary wicking (Figure S6)" in the revised manuscript. In addition, corresponding descriptions are added in the figure caption of Figure S6 in the revised Supplementary Information.

- 2. 7. higher structure temperature > need other expression**

Additional suggestion: The expression "higher structure temperature" is somehow

confusing and may be understood as the temperature of the higher structure. Please rephrase the expression to avoid confusion. Higher structure temperature -> Higher temperature of the structure.

Reply: Thanks for the reviewer's suggestion. To avoid the confusion of the expression, we have revised the "higher structure temperature" to "higher temperature of the apex structure" in Line 22, Page 7 in the revised manuscript.

3. 14. "Before use, the front side of 3D printed structures were plasma treated to endow the upper surface with hydrophilicity for water spreading." For better understandings of hydrophilicity, please supply XPS result for upper surface in supporting information.

Additional question: What is the meaning of the "peak value of the surface oxygen"? Do the authors imply the height or area of O1s peak? And if so, how can those increases be evidence of the increases of the -OH groups? It is advisable to provide proper peak deconvolution results to clearly show changes in the surface chemistry. Furthermore, it will give more clarity if C1s peak is provided and sufficiently analyzed as well.

Reply: Thanks for the reviewer's correction. We have compared the C 1s peak and the O 1s peaks before and after plasma treatment, as displayed in **Figure R1** (**Figure S18** in the revised supplementary information). The atomic ratio of oxygen increases from 21.7% to 33.11% after plasma treatment, which means that the number of oxygen-containing groups rises. In addition, the C 1s and O 1s spectra showed increase in the surface concentration of O=C=O functional groups as well as enrichment of C-O groups on the structure's surface after plasma treatment, which contributes fundamentally to the improvement of surface hydrophilicity¹.

Accordingly, we have revised the description of the XPS result in the **Methods** part of the revised manuscript and the Figure caption of Figure S18 in the revised Supplementary Information.

Figure R1 (Figure S18 in the revised Supplementary Information). a. XPS spectra of C 1s peaks of the upper surface of the bio-mimetic 3D evaporator before (black line) and after (red line) plasma

treatment. C1 at a binding energy of 284.8 eV corresponds to the C*-C* and C*-H bond. C2 at 286.4 eV reflects the C*-O bond, and C3 at 288.9 eV corresponds to O=C*-O bond. **b.** XPS spectra of O 1s peaks of the upper surface of the bio-mimetic 3D evaporator before (black line) and after (red line) plasma treatment. O1 at a binding energy of 532.2 eV corresponds to C=O* bond and O2 at 533.6 eV represents O*-C=O bond.

Responses to Reviewer # 2

All comments from reviewers have been addressed carefully. The authors have also conducted additional experiments and simulation to support their conclusions. I would like to recommend a publication in NC although a few minor issues are still needed to be addressed:

Reply: The authors thank the reviewer very much for the positive comments and the additional suggestions for improving the manuscript. According to the reviewer's comments, we have made revisions on our manuscript and addressed them one by one as below. The suggestions improve the quality of the revised manuscript. Thanks again for the reviewer's comments and suggestions.

1. More details about the stimulation are needed to be provided in the supporting materials, such as the mesh, boundary conditions, physical parameter and validation.

Reply: Thanks for the reviewer's suggestion. We have added the detailed mesh conditions and boundary conditions used for the simulation process in **Figure S19** (Figure R2) with corresponding descriptions in the figure caption in the revised Supplementary Information. We have also added the physical parameters used in the simulation process in **Table S3** (Table R1) and **Figure 20** (Figure R3) in the revised Supplementary Information. The validation had been in the original manuscript as "Numerical simulation is further employed to prove the existence of the position-related water evaporation phenomenon, the result of which is consistence with the above analysis, where the steam diffusion flux at the apex is larger than that at the bottom under one sun illumination as shown in Figure 3f" in Line 12, Page 8. Combing with the **Numerical Simulation** section in the **Methods** part of the revised manuscript, the simulation process is thus repeatable.

Figure R2 (Figure S19 in the revised Supplementary Information). Detailed geometrical conditions of the simulation. a. Scheme of the model used for numerical simulation of the wet

bio-mimetic 3D structure in darkness and under one sun illumination. Because of the symmetry of the structure, one groove of the wet bio-mimetic structure in darkness and under one sun illumination is simulated. **b.** Mesh conditions of the simulation model. The maximum mesh size is 1 mm and the minimum of the mesh size is 0.05 mm. Mesh is refined at the boundaries of water film surface with the highest resolution. **c.** The red boundary represents the boundary of the ambient environment and is set as 293.15 K and relative humidity 0.5. **d.** The yellow boundary represents the water film surface and is set as relative humidity 0.5 and latent heat source by evaporation. **e.** The blue boundary which represents the boundary of the interface of the 3D structure and the water film. Under one sun illumination, it exists as the heat source of 0.5 W/m^2 , while in darkness, it exists as not energy output. Other undefined boundaries are set as zero flux of energy or steam.

Table R1 (Table S3 in the revised Supplementary Information). Material parameters for the numerical simulation process.

Material properties	Symbol	Value
heat conductivity coefficient of water	k_1	$0.59 \text{ W/(m}^*\text{K)}$
heat conductivity coefficient of air	k_2	$0.026 \text{ W/(m}^*\text{K)}$
heat conductivity coefficient of the 3D evaporator	k_3	$0.19 \text{ W/(m}^*\text{K)}$
diffusion coefficient of vapor	D	$2.6\text{E-}5 \text{ m}^2/\text{s}$
saturated vapor concentration*	c_{sat}	$0.8\text{-}1.6 \text{ mol/m}^3$
latent heat coefficient	L_v	$2.4\text{E}6 \text{ J/Kg}$

*The saturated vapor concentration is temperature-dependent, the plot of saturated vapor concentration with temperature is shown in Figure S20.

Figure R3 (Figure S20 in the revised Supplementary Information). The plot of saturated vapor concentration with variation of temperature.

2. The evaporation rates reported in Table R1 and Table S2 are different. Please double check.

Reply: Thanks for the reviewer’s comment. We are sorry for the mistake and we have corrected the Table S2 in the revised supplementary information.

3. In the section of “The generation of the temperature gradient under one sun illumination”, the authors state, “Due to the position-related structure inhomogeneity, the energy that the dry 3D structure can acquire from the illuminated light is different along the sidewall. With a closer position from the light source, higher energy intensity can be acquired at the apex. Cooperating with a smaller surface area at the apex, it will lead to a higher apex structure temperature.”

It is true for a solar simulator. However, in real situations, this distance-related energy intensity may be negligible. In addition, what do you mean by “the apex has smaller surface” and why it will lead to a higher apex structure temperature? Please give more description.

Reply: Thanks for the reviewer’s comments. In real situations, this distance-related energy intensity may be negligible for 2D evaporators, but cannot be negligible for 3D evaporators (Joule, 2019, 3, 1798-1803, Ref. 44 in the manuscript). In order to normalize the energy that the whole 3D structures with different heights can acquire, we used the top projected plane of 3D evaporators as the calibrating plane for measuring the light intensity, as shown in **Figure R4**. The light intensity is equal to one sun at the top projected plane of corresponding 3D structures. However, energy acquisition from the illuminated light along the sidewall for 3D evaporators with different H/D ratios is different, which is still position-related ascribing from the different structure inhomogeneity of 3D structures with different H/D ratios. As displayed in Figure 3b in the manuscript, the temperature

profile of the dry structure before water coverage, the temperature difference along the sidewall is obvious, which experimentally proves the position related energy acquisition.

Figure R4. Scheme of the top projected area where the illuminated light intensity is equal to one sun.

We are sorry for the wrong utilization of the phrase “small surface area”. We want to demonstrate that the apex structure has a large specific surface area (surface to volume ratio). As displayed in the yellow shadow covered region, the heights of the yellow shadows are set as H. The specific surface area of the apex position and bottom position can be calculated from the parameters displayed in Figure S2a. The apex position possesses a much larger specific surface area than that at the bottom. Under the same light illumination, the apex position with a higher position and a larger specific surface area acquires more energy than that at the bottom.

Accordingly, we have revised the sentence “Cooperating with a smaller surface area at the apex, it will lead to a higher apex structure temperature” to “Cooperating with a larger specific surface area at the apex, it will lead to a higher temperature of the apex structure” in Line 21, Page 7 the revised manuscript.

References

1. Vesel, A., Mozetic, M. & Zalar, A. XPS study of oxygen plasma activated PET. *Vacuum* **82**, 248-251 (2007).